# OmniCode: A Benchmark for Evaluating Software Development Agents

## Abstract

LLM-powered coding agents are redefining how real-world software is developed. To drive the research towards better coding agents, we require challenging benchmarks that can rigorously evaluate the ability of such agents to perform various software engineering tasks. However, popular coding benchmarks such as HumanEval and SWE-Bench focus on narrowly scoped tasks such as competition programming and patch generation. In reality, software engineers have to handle a broader set of tasks for real-world software development. To address this gap, we propose OmniCode, a novel software engineering benchmark that contains a diverse set of task categories, including responding to code reviews, test generation, fixing style violations, and program repair. Overall, OmniCode contains 1,794 tasks spanning three programming languages—Python, Java, and C++—and four key categories: bug fixing, test generation, code review fixing, and style fixing. In contrast to prior software engineering benchmarks, the tasks in OmniCode are (1) manually validated to eliminate ill-defined problems, and (2) synthetically crafted or recently curated to avoid data leakage issues, presenting a new framework for synthetically generating diverse software tasks from limited real world data. We evaluate OmniCode with popular agent frameworks such as SWE-Agent and show that while they may perform well on BugFixing, they fall short on tasks such as Test Generation and in languages such as C++. OmniCode aims to serve as a platform for generating synthetic tasks from real world data, spurring the development of agents that can perform well across different aspects of software development.

## 1 Introduction

The future impact of AI-automated software development will be far-ranging: beyond building and improving apps, AI will help us write more comprehensive test suites, perform and respond to code review suggestions, enforce nuanced style guidelines, and perform many other tasks that are part of the software development life cycle. Research on AI software development demands good benchmarks, both to measure progress and to expand the scope of problem statements. However, AI coding benchmarks today, such as SWE-Bench (Jimenez et al., 2024), CodeContests (Li et al., 2022), and HumanEval (Chen et al., 2021), are too narrow in scope to spur progress on automating the full spectrum of software development tasks, instead focusing on isolated tasks such as competition programming, code repair, and generating individual patches in isolation.

**OmniCode.** To address this gap, we introduce a new benchmark for generative AI coding assistants (specifically LLMs for code), which we call OmniCode. Our new benchmark is based on the insight that software development involves a heterogeneous range of tasks and problem-solving activities for which generative AI can be brought to bear (see Figure 1). We consider four such software development tasks:

Figure 1: Omnicode synthetically builds multiple tasks out of a base dataset to holistically evaluate software engineering agents. Four different types of tasks that we consider: Bug fixing/feature adding, test generation, responding to code review, and enforcing style guidelines.

1. Addressing issues, such as bug fixes and feature requests. This is a staple of software engineering benchmarks (Jimenez et al., 2024; Silva & Monperrus, 2024; Rashid et al., 2025), because it assesses the ability of an LLM coding agent to autonomously resolve real-world repository-level issues, provided we are given tests for validating program correctness.

2. Writing software tests. Current LLM coding agents are unreliable, requiring humans to manually inspect and test their outputs. By having LLM coding systems write their own tests, we measure progress toward fully closing the loop of both generating and checking repo-level patches.

3. Responding to code review. Coding agents today act in a partnership with human engineers, and we envision a future where LLMs provide initial drafts of a patch, which a human engineer then critiques. We compile a dataset of partly-correct patches paired with code-review feedback on how to best correct them, and task models with completing or fixing the patch given the code review.

4. Enforcing style guidelines. Code style is important for conforming to project-specific or organization-specific norms. Here, present the agent with a selection of coding convention violations in a file and test the ability of an LLM to fix the style.

We build our benchmark by bootstrapping off existing benchmarks such as SWE-Bench and Multi-SWE-Bench, along with collecting additional issues from popular open-source repositories. Using this collected real-world data, we employ LLM-based augmentations along with language-specific tools to create different task types. In total, our dataset comprises 494 issues from 27 repositories and 1794 benchmark tasks in total.

**Results.** We evaluate the widely used SWE-Agent with models spanning a range of providers and sizes (Gemini 2.5 Flash, DeepSeek-V3.1, GPT5-mini and Qwen3-32B) on our dataset. We also evaluate Aider Aider-AI (2025) with Gemini 2.5 Flash as pipeline-based agent comparison to SWE-Agent. We find that our benchmark challenges even the most modern systems, but it is not intractable. Specifically, SWE-Agent achieves a maximum of 25% on test generation across all three languages. On Review-Response it achieves a maximum of 52% on Python. For Style-Fixing, while agents perform well on Python, they do not perform as well on Java and C++. We also observe significant variations between models and agent frameworks.

**Contributions.** We wish to highlight the following contributions:

1. OmniCode, a benchmark assessing for distinct types of software engineering activities, comprising 1794 tasks total.

2. Presenting recipes for synthetically creating diverse interactive tasks to evaluate agents from collected static real-world data.

3. Empirical evaluation of state-of-the-art LLM-agent systems on the benchmark, determining specific areas where LLM agents fall especially short, particularly in test generation and style fixing.

## 2 RELATED WORK

**LLM coding benchmarks.** One of the earliest benchmarks for LLMs' functional code synthesis was HumanEval (Chen et al., 2021), which contained 164 hand-written programming problems, each with a natural language docstring and associated unit tests. However, it was limited to single-function synthesis without any multi-file or repository context. SWE-Bench Jimenez et al. (2024) first introduced the paradigm of benchmarking the ability of LLM agents to resolve real-world GitHub issues, yielding much follow-up work (Miserendino et al., 2025; Jain et al.; Aleithan et al., 2024; Rashid et al., 2025; Zan et al., 2024). These benchmarks added support for more languages and improved data quality by including more rigorous checks. Similar to these benchmarks, we also manually validate each base task before including it in OmniCode. In contrast to these benchmarks, OmniCode contains three new synthetic tasks that reduce the chances of data leakage. Recently SWE-Smith Yang et al. (2025) has shown promise in synthetically generating bugs to create training data for coding agents. OmniCode goes beyond just new bugs, to creating new task types that are supported by synthetically generated data, such as code reviews. Multi-SWE-Bench Zan et al. (2025) extended the SWE-Bench collection paradigm beyond Python to multiple languages, but restricted to bug-fixing. We further extend this to other tasks that are part of the software development process.

**LLM coding benchmarks for other tasks.** Recently, Mündler et al. proposed SWT-Bench (Mündler et al., 2024) that transforms the instances in SWE-Bench to test generation tasks. Each task involves generating tests such that they fail on the buggy version of code and pass with the fixed version e.g, the gold patch, which we call Fail-to-Pass. In contrast to SWT-Bench, the test generation tasks in OmniCode are more robust. Our tasks require not only the generated test to go from Fail-to-Pass for golden patch but also Fail-to-Fail when presented with multiple bad patches requiring the agents to generate tests that don't pass trivially, resulting in more robust tests. TestEval (Wang et al., 2024) is another recent benchmark for evaluating test generation capabilities of LLMs. However, their benchmark is only set up for single programs instead of entire repositories, which is more challenging.

**Test case generation with LLMs.** Past work has also built LLM program synthesizers organized around the principle of self-checking through test case generation (Li et al.; Chen et al., 2022). Researchers have also proposed generating unit tests using LLMs (Chen et al., 2024; Pan et al., 2024). However, these works are either focused on using tests as a validation step or improving unit test generation for a given focal method with a single LLM. In contrast, our work focuses on benchmarking LLM-Agents for repository-level test generation.

## 3 BENCHMARK CONSTRUCTION

The creation of OmniCode involves two major steps: (1) gathering real-world software data from open source repositories and (2) generating augmentations on these base instances to support new task types. Each instance in our benchmark is based on a pull request that has been made to resolve an issue in a GitHub repository. The pull request and its associated metadata (such as the issue it resolved, the patch it introduced)

constitute what we call a base instance. Using this base instance, we can generate the data required to support different task types, such as generating bad patches to support test generation or code reviews to support review fixing. Next, we describe both the data collection and task generation in detail.

## 3.1 COLLECTION OF REAL-WORLD DATA FROM GITHUB

We first collect a set of base instances, that is, pull requests in public GitHub repositories, from which we can generate tasks. When curating pull requests, we follow a similar selection strategy to Jimenez et al. (2024). We consider popular projects, filtering out tutorials and other non-code repositories. From these, we collect merged pull requests that (1) resolve an issue and (2) introduce a test. To ensure that each instance is a meaningful task for an agent to be evaluated on, we perform manual inspection. Only instances where the changes introduced in the pull request are within the scope of the description of the issue are kept. We also discard issues if they only involve trivial changes to documentation or configuration files.

To enable agents to interact with an instance by executing code, we build containerized environments for each instance. The environment is made up of the state of the repository at the time of the issue, as well as dependencies that need to be installed so that code can be executed properly. We manually determine the dependencies required by inspecting requirements and documentation. To verify that the correct dependencies have been identified, we execute the test suite of the repository to check if the tests can be run without errors.

For our evaluation, we curate a multi-language dataset by filtering and selecting sane and reliable instances from existing benchmarks such as SWE-Bench and Multi-SWE-Bench, and we supplement this with a small number of additional repositories and hand-picked instances. This combined dataset comprises 273 Python, 112 C++, and 109 Java instances (494 in total), spanning 28 diverse repositories across machine learning and scientific libraries (e.g., scikit-learn, sympy), systems libraries (e.g., fmt, simdjson), and large-scale frameworks (e.g., django, logstash, jackson, mockito). By extending coverage to Java and C++ in addition to Python, our dataset broadens evaluation beyond the Python-centric scope of SWE-Bench, providing a more realistic and comprehensive benchmark for assessing software engineering agents across ecosystems.

## 3.2 TASK DETAILS

In the following, we describe the details of how each of our main four task types is set up along with the evaluation procedures.

### 3.2.1 TASK: RESOLVING ISSUES

Resolving GitHub issues has become a standard approach for evaluating the capabilities of large language models (LLMs) in the software engineering domain. A common method, first introduced by Jimenez et al. (2024) is to mine resolved issues from large-scale open-source repositories. This provides a natural environment for agents to operate in by cloning the corresponding repository state, including the issue description, and withholding a set of tests used to validate the proposed fix. For each instance, we provide the issue description and a set of tests that distinguish between the pre- and post-fix repository states. An agent is tasked with generating a patch based on the issue, which is evaluated against tests that transitioned from failing to passing due to the ground truth fix, as well as against previously passing tests to ensure no regressions are introduced. While this task aligns closely with existing work, our benchmark expands the range of verified projects considered to by unifying instances from SWE-Bench, Multi-SWE-Bench, as well as 37 instances that we collect while maintaining a strong emphasis on manual validation for quality assurance.

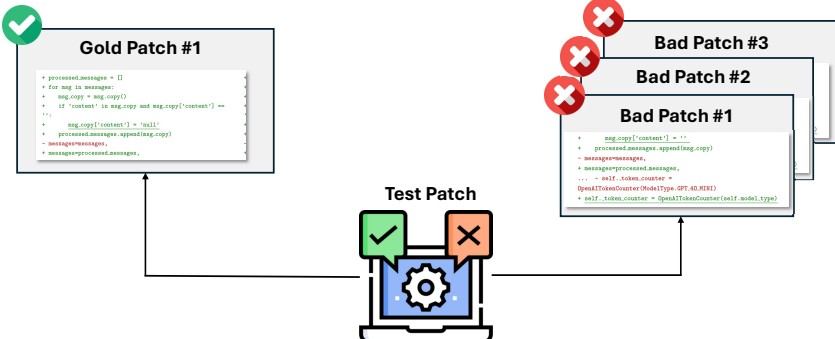

Figure 2: For evaluating test patches on the task of Test Generation, we evaluate the proposed test patch against both the ground truth (gold) patch, as well as several meaningful, but incorrect, bad patches. A test is only considered correct if it passes for the gold test, but fails for all bad patches.

### 3.2.2 TASK: TEST GENERATION

All previously considered pull requests included relevant tests, as this was a necessary criterion for their selection. These tests play a crucial role in verifying that the proposed fix is valid and addresses the reported issue. However, this requirement significantly limits the number of available instances for model evaluation. At the same time, writing meaningful tests is itself a key aspect of software engineering. By focusing on this underexplored skill, we aim to evaluate and improve a model's ability to reason about code behavior and generate effective test cases.

To assess the quality of a candidate test, we use both the ground truth test case and a set of what we define as bad patches. A bad patch is a plausible but incorrect attempt at resolving the issue—one that contains no obvious syntax errors and remains relevant to the problem description. This setup presents a more realistic and challenging evaluation scenario compared to existing approaches, which typically rely only on the pre- and post-PR repository states.

While there are usually few ways to correctly solve a problem, there are many ways to incorrectly solve it. To ensure that generated tests can be evaluated thoroughly, it is important to have bad patches that cover a diverse set of failure modes. We use two distinct approaches to achieve this. (1) Collecting failed attempts from less capable agents and (2) Perturbing correct patches to introduce bugs. For approach (1), we use Agentless (Xia et al., 2024) with several different models (Gemma 2 9B, Qwen2.5 Coder 32B Instruct, Llama 3 8B Instruct, and GPT-4.1-nano), instructing the tool to attempt to solve the task as usual and collecting instances where it fails to do so. For approach (2), we sample multiple completions from Gemini 2.0 Flash, prompted with the correct patch along with instructions to perturb it in order to introduce commonly found bugs, filtering to keep those that are actually incorrect. The relevant prompt can be found in the appendix. Our aim is to have bad patches which are incorrect in minor ways (from approach 2) as well as at a higher level (from approach 1).

For the Java and C++ instances, we placed more emphasis on the Agentless generations for their more natural patch attempts. However, there were instances that proved to be resilient to bad patch generation. These were instances that either proved too difficult for the models to produce a valid patch or too simple for them to produce a non-passing patch. As a result, we were limited to a subset of our instances for Java and C++. For Java, we used 77 instances for this subset. For C++, we used 44 instances for this subset.

In this setting, the agent is prompted with the issue text and asked to generate one or more test cases to be added to the test suite. The resulting candidate test is then evaluated: if it passes on the ground truth patch but

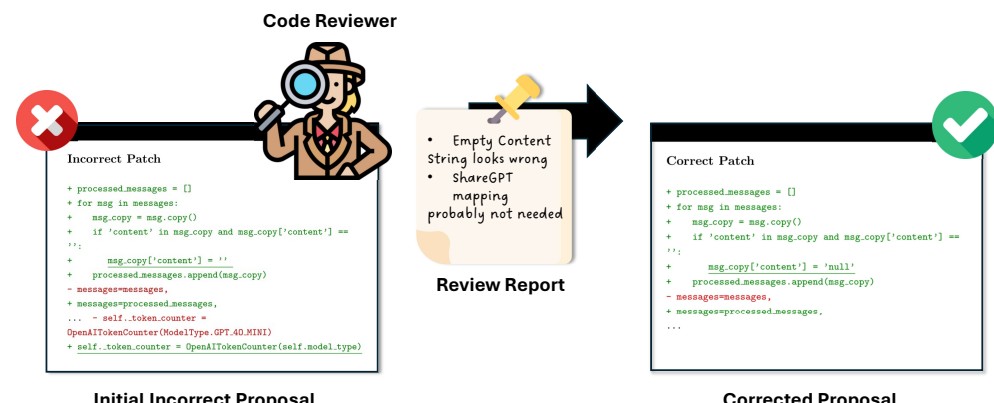

Figure 3: In the task of responding to Code Review, an initial incorrect patch is provided, which contains a meaningful attempt of the solution of a given problem. This attempt is then reviewed by a human or an LLM, and a review report is generated. Utilizing this report, the LLM is tasked with correcting the initial approach by utilizing this report, which is validated with the normal testing suite.

fails on all bad patches, it is considered successful. If it does not meet both criteria, the test is considered a failure. We also reuse the bad patches in an additional task related to code review.

### 3.2.3 TASK: RESPONDING TO CODE REVIEW

It is not uncommon for developers to iterate over multiple proposed solutions in a pull request until they fulfill all the necessary requirements. Often, such incorrect proposals are met with corresponding feedback or review, explaining why or how this approach does not meet expectations. We create reviews by providing both the perturbed bad patch (from the previous section) along with the correct patch and problem description to Gemini 2.0 Flash, and asking it to come up with instructions for how the bad patch should be fixed. We create our prompt in order to induce reviews that are informative but do not give away the complete solutions.

During evaluation, we present the model with the previously selected bad patch and display the review of context. The model is then tasked with refining the existing solution in a way that passes the issue-specific fail-to-pass test. While the adaptation of existing functionality to enable this use case is minor, we believe this is a promising avenue for research. Especially when anticipating fully autonomous work on code issues, interacting with external feedback, and starting from potentially corrupted states is an imperative skill.

### 3.2.4 TASK: CODE STYLE

Last, we introduce the task of style review. Since language models are trained on a wide range of code—varying not only in functionality but also in quality—style-oriented tasks represent a natural extension of evaluation. To assess code style, we use third-party tools such as `pylint` for Python, `clang-tidy` for C++, and `PMD` for Java to score quality and extract specific style issues, including errors, warnings, and convention violations.

In this task, the model is not expected to fix a functional bug but to resolve the listed style issues. Style review is particularly appealing because it can be adapted to user-specific needs by incorporating custom guidelines or organization-specific rules.

**Code Style Review**

| Before | Linter Report | After |
|--------|---------------|-------|
| ```def is_pos_difference(...): ... difference = a - b is_pos = difference > 0 return is_pos``` | ```{ "type": "refactor", ... "message": "Too many local variables", ... }``` | ```def is_pos_difference(...): ... return a > b``` |

Figure 4: Side-by-side display of the original verbose code, linter warning, and refactored code with reduced local variables. Key elements highlighted in blue.

We construct datasets for style errors for all repositories used for other tasks. We start by using the language-specific tools to generate a list of all style violations in the repository. We then aggressively prune out overly zealous rules and other commonly occurring warnings. We record both an aggregate style score and the full list of reported style issues, including line numbers. We then group errors by file and construct 144 Python, 147 C++, and 124 Java instances, with each instance containing on average 9 style errors.

This output is passed to the agent, which is then tasked with resolving the identified issues. After applying the proposed patch, we re-run the style tool and quantify improvement based on score increase or the number of issues eliminated. To account for partial success, we allow a relaxed pass criterion, configurable via thresholds on minimum score or maximum remaining issues. To determine how well the agent resolve style violations, we compute a metric using a the following formula that balances the total number of instances resolved with new ones that are introduced, normalizing by total number of issues initial present:

$$\text{score} = \max\left(\frac{\text{resolved} - \text{new}}{\text{original}}, 0\right)$$

### 3.3 EXPERIMENTAL SETUP

To demonstrate our benchmark, we evaluate the state-of-the-art agent framework SWE-Agent, along with a more pipelined and less agentic approach: Aider. We evaluate both frameworks with Gemini 2.5 Flash. In order to enable agents to interact with the instances, we provide them with containerized environments as described in Section 3.1. We pass in the issue description as the initial task statement for Bug-Fixing. For Test-Generation, Review-Response, and Style-Fixing, we prepare task-specific prompts that provide context and instructions. These are detailed in the appendix. We use the default settings for SWE-Agent and adjust the per instance cost limit to $2.0.

Table 1: Combined statistics by language

| Metric | Python | C++ | Java |
|--------|--------|-----|------|
| *Patch statistics* | | | |
| Patches | 273 | 112 | 109 |
| Complexity | 7.1 | 47.6 | 19.2 |
| Lines added | 16.9 | 180.7 | 74.8 |
| Lines removed | 7.7 | 82.6 | 20.3 |
| *Test statistics* | | | |
| Patches | 273 | 112 | 109 |
| Complexity | 7.2 | 38.0 | 11.9 |
| Lines added | 25.2 | 277.8 | 72.2 |
| Lines removed | 4.9 | 17.5 | 2.0 |
| *Bad Patch and Review statistics* | | | |
| Patches | 164 | 44 | 79 |
| Complexity | 2.870 | 3.641 | 3.056 |
| Lines added | 3.909 | 5.455 | 5.785 |
| Lines removed | 1.866 | 2.318 | 1.861 |
| Review size | 253.6 | 319.6 | 329.0 |

## 4 ANALYSIS OF DATASET

**Bug Fixing** In Table 1, we present quantitative analysis of the patches that introduce the bug into the repository. Along with the size of patches, we construct a metric to better gauge bug complexity as complexity = $\Delta$Files + Hunks + (AddedLines + RemovedLines)/10. We observe that the tasks follow difficulty order by language as C++ > Java > Python. We see that this is reflected in the performance of agents on the tasks too.

**Test Generation** In Table 1, we present a similar analysis for test patches, quantifying the complexity of the tests that need to be generated in the Test Generation task. We observe that the tasks follow the same difficulty order by language as for BugFixing: C++ > Java > Python.

**Review Response** In Table 1, we also present an analysis of bad patches generated using Agentless, along with sizes of Reviews generated for these bad patches, observing similar trends for

## 5 ANALYSIS OF LLM CODING AGENTS ON OMNICODE

Table 2: SWE-Agent Performance across languages and models

| Language | Model | Bug-Fixing | Test-Generation | Review-Response | Style-Fixing |
|---|---|---|---|---|---|
| Python | Gemini-2.5-Flash | 38.1% | 14.0% | 29.9% | 72.2% |
| | DeepSeek-V3.1 | 56.4% | 18.7% | 52.2% | 73.4% |
| | GPT-5-mini | 47.3% | 6.2% | 30.5% | 56.3% |
| | Qwen3-32B | 24.5% | 4.0% | 17.7% | 22.7% |
| C++ | Gemini-2.5-Flash | 8.0% | 12.2% | 13.6% | 36.3% |
| | DeepSeek-V3.1 | 19.6% | 25.0% | 22.7% | 30.2% |
| | GPT-5-mini | 15.2% | 6.8% | 20.5% | 21.8% |
| | Qwen3-32B | 3.8% | 4.5% | 4.5% | 8.6% |
| Java | Gemini-2.5-Flash | 14.7% | 4.9% | 31.6% | 60.4% |
| | DeepSeek-V3.1 | 31.2% | 20.9% | 44.3% | 50.2% |
| | GPT-5-mini | 22.0% | 2.7% | 26.6% | 25.0% |
| | Qwen3-32B | 10.1% | 1.3% | 15.2% | 23.3% |

Table 3: SWE-Agent vs Aider Comparison

| Language | Agent | Bug-Fixing | Test-Generation | Review-Response | Style-Fixing |
|---|---|---|---|---|---|
| Python | SWE-Agent | 38.1% | 14.0% | 29.9% | 72.2% |
| | Aider | 32.4% | 9.4% | 26.8% | 60.3% |
| C++ | SWE-Agent | 8.0% | 12.2% | 13.6% | 36.3% |
| | Aider | 1.8% | 2.3% | 4.5% | 10.1% |
| Java | SWE-Agent | 14.7% | 4.9% | 31.6% | 60.4% |
| | Aider | 19.3% | 3.9% | 25.3% | 60.9% |

### 5.1 PERFORMANCE ACROSS TASKS

We present the results of evaluating SWE-Agent with a range of state of the art LLMs in Table 9. We find that while a state of the art system like SWE-Agent excels on some tasks like Style Fixing in python, there are many holes in its abilities. Specifically, we observe that it struggles at Test-Generation, where all tools

struggle across languages, the maximum performance being 25% on Python. Test generation is an essential skill for SWE Agents for (1) assisting humans in developing robust test suites but also (2) writing tests to verify their own code is correct. The evaluated tools also suffer disproportionately at C++, which agrees with our analysis in Section 4 regarding the complexity of C++ bugs over other bugs in our benchmark. We find that when using SWE-Agent, performance of different models on bug-fixing is strongly correlated to review-response (pearson coeff = 0.921) and weakly correlated to test generation (pearson coeff = 0.764). We find the correlation does not hold for style-review however (perason coeff = 0.512), where Gemini-2.5-Flash performs as good as or better than DeepSeek v3.1 on Style-Fix despite DeepSeek consistently outperforming Gemini on Bug-Fix. We find these observations to be generally true for Aider too, albeit slightly weaker. Details of correlation analysis are presented in the Apendidx E.

## 5.2 COMPARISON BETWEEN AGENTS

We compare a widely used agentic approach (*SWE-Agent*) with a pipeline-based approach (*Aider*) to assess the strengths and weaknesses of both paradigms. As shown in Table 3, *SWE-Agent* consistently outperforms Aider across most programming languages and task types when evaluated on OmniCode using Gemini-2.5-Flash. For Python, *SWE-Agent* achieves higher performance in bug-fixing (36.7% vs. 32.4%), test-generation (14.0% vs. 9.4%), and review-response (29.9% vs. 26.8%), reflecting its stronger reasoning and synthesis capabilities. In C++, *Aider* performs substantially worse, while *SWE-Agent* maintains modest but consistent gains, particularly in test-generation (12.2% vs. 2.3%) and review-response (13.6% vs. 4.5%). One possible explanation is that C++ tasks in OmniCode require more interactive reasoning and iterative error analysis, involving multiple compile-run cycles and complex dependency handling. *Aider's* pipeline-oriented design may thus struggle with such trial-and-error-intensive workflows. Overall, these findings indicate that while *Aider* remains competitive on less interactive or simpler tasks, SWE-Agent demonstrates greater robustness and adaptability to complex, multi-stage software engineering problems, particularly those requiring sustained reasoning and feedback integration. These results highlight OmniCode's ability to differentiate between interaction-intensive and procedural tasks, providing a nuanced view of how agentic and pipeline systems handle varying levels of task complexity and reasoning demand.

## 5.3 REVIEW-RESPONSE

It is a well-known challenge for language models to identify the correct entry point when resolving issues in large, multi-file repositories. We hypothesized that providing structured feedback via a Review-Response task would improve performance over an autonomous Bug-Fixing task by guiding the agent. To test this, we benchmarked several LLMs (including Gemini-2.5-Flash, DeepSeek-V3.1, GPT-5-mini, and Qwen3-32B) across Python, Java, and C++. As all models showed a strong positive correlation (Table 9), we focus our analysis on the results from DeepSeek-V3.1. While overall performance varied by language (Python > Java > C++), the analysis supports our hypothesis that the guided Review-Response task is a more effective problem framing. Since all Review-Response instances are a subset of Bug-Fixing, we can directly compare performance on this common set. Here, Review-Response consistently resolved more unique instances: for Java, it uniquely resolved 15 instances versus Bug-Fixing's 4, a pattern that held for C++ (4 vs. 2) and Python (22 vs. 20). The seemingly contradictory raw scores for Python (56.4% Bug-Fixing vs. 52.2% Review-Response) are explained by the non-review instances being comparatively easier, with a high 65.1% resolution rate. We also investigated common failure modes. Java, for instance, was most susceptible to producing empty patches (8.9% in Review-Response vs. 6.4% in Bug-Fixing).

## 5.4 PATCH COMPLEXITY

As shown in Figure 12, the complexity score distribution for unresolved instances is significantly higher than that of the resolved ones, which reveals a negative correlation between successful resolution and patch complexity. For details, refer to Table 10 and Table 11 in appendix. We further investigated the structural complexity of generated patches to understand how agents approach different languages. The ground truth (Gold) patch complexity followed a clear hierarchy: C++ (47.55) > Java (19.24) > Python (7.07). DeepSeek-V3.1 demonstrated the highest stability, maintaining generation complexity closest to the Gold standard, whereas other models exhibited a tendency toward "explosive" complexity in unresolved instances. For example, GPT-5-mini's unresolved Python patches reached an average complexity score of 390.18 - far exceeding the Gold average of 7.07. We hypothesize this happens when the agent is unable to pinpoint a precise fix, it attempt sprawling, ineffective refactors. Conversely, successful resolutions were often "cleaner" than human-written solutions; for instance, DeepSeek's resolved Python patches averaged a complexity of 5.35 compared to the Gold 7.07. Notably, the Review-Response framing did not effectively constrain this volatility, as complexity scores for unresolved patches remained unstable or even increased. Unlike the explosive failures in Python, unresolved Java patches consistently retained low complexity (e.g., Qwen3-32B averaged 5.08 vs. Gold 19.24), suggesting that the language's strict syntax discourages the refactoring seen in more dynamic languages.

## 5.5 Impact of including Bad Patches

Incorporating bad patches is essential for evaluating the true robustness and discriminative power of LLM-generated test cases. Metrics based solely on gold-patch success (as in prior work) dramatically overestimate a model's testing capability. In analysis of success for Qwen and DeepSeek results, test cases would have been accepted at a higher rate if bad-patch failures were not required (e.g., Qwen C++ would be 22.7% instead of 4.55%, Qwen Java would be 7.79% instead of 1.3%, DeepSeek C++ would be 43.8% instead of 25%, and DeepSeek Java would be 28.4% instead of 11.9%). This gap highlights that many generated tests capture superficial behaviors rather than the underlying program semantics. By enforcing that gold patches pass and all bad patches fail, we obtain a far more realistic assessment of test quality, one that reflects a model's ability to differentiate correct logic from subtly incorrect implementations, a critical requirement for trustworthy automated testing.

## 6 Limitations and Future Work

Although we believe that our work expands the extent to which LLM coding benchmarks span the spectrum of software engineering activities, much remains to be done before we truly have a test of whether an AI system can perform as a programmer. Real programmers deal with config files, multiple languages, profiling and optimizing, and engage in natural language conversation to iron out design decisions, plan sprints, and other forms of team strategy. Although our benchmark is a step toward a more comprehensive assessment of these systems, further expanding the suite of heterogeneous software-engineering tasks remains a prime target for future research. We are currently working on expanding OmniCode to 1) other languages beyond Python, Java, and C++, and 2) additional task categories like fixing security violations and code migration. Both of these are emergent fields which we aim to adapt as soon as possible. Transitioning functionality between languages is a very challenging, but fruitful tasks, which has seen only little attention in the evaluation field of large language models. Similarly, spotting and fixing security violations requires a very deep understanding of system dynamics, which language models may not yet possess. Further, specific tool usage, as is needed for tasks like style review, carries over naturally to other programming languages. Our implementation already employs checkstyle as a java-based alternative to pylint, in order to enable style review for repositories of both origins. We believe that expanding the diversity of tasks and languages in this way will enable a more robust evaluation for LLMs and LLM-Agents.

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

## A  ANALYSIS OF BAD PATCHES AND REVIEWS

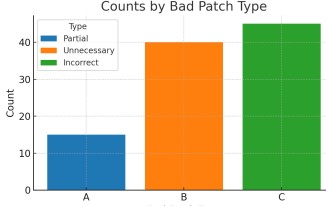

Figure 5: Categorization of bad patches.

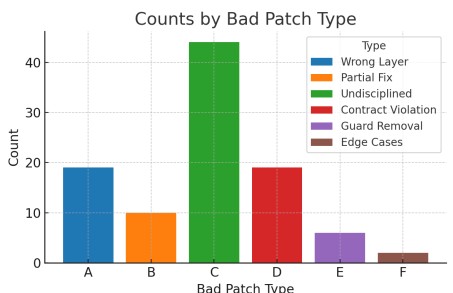

Figure 6: Categorization of bad patches

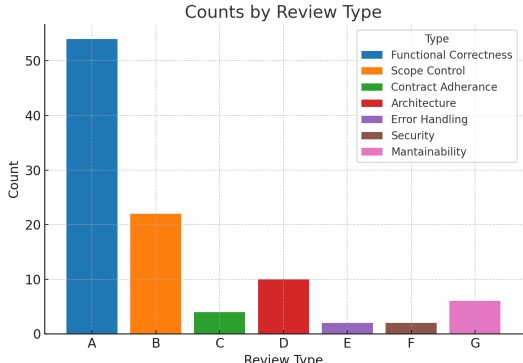

Figure 7: Categorization of reviews

To understand the distribution of bad patches generated by our pipeline we categorise a sample of 100 python bad patches with results displaying in Fig 5. The categorisation is performed by prompting an LLM with descriptions of the category along with the problem description, bad patch and correct patch for the instance.

We observe that bad patches are distributed across a range of types, with most of them being "Undisciplined", that is patches which make more spurious changes than necessary. There are also a significant number of bad patches in the "Wrong Layer " and "Contract Violation" categories.

Another way to understand bad patches is to categorise them according to whether they are "Partial" (the attempted fix is partially correct), "Unnecessary" (the patch makes spurious changes) or "Incorrect" (the fix approach is incorrect). We observe that the majority of bad patches are due to incorrect approach at making the fix. These patches are useful to include in the dataset as they characterise probable failure modes that existing tests may not account for.

To understand the distribution of reviews generated by our pipeline we categorise a sample of 100 python reviews with results displaying in Fig 3. The categorisation is performed by prompting an LLM with descriptions of the category along with the problem description, bad patch, correct patch and review for the instance.

We observe that the vast majority of reviews are to do with improving functional correctness. There are also reviews that discuss "Scope Control" and "Architecture".

Descriptions used to categorise bad patches -

**A. Wrong-layer fix / misdiagnosed root cause**
*Description:* The change targets the wrong component or symptom instead of the source of truth. Signals include modifying outputs instead of inputs, tweaking helpers when call sites or flags need changes, relying on attributes/settings that are never wired, or making comment-only/no-op changes.

**B. Partial fix / incomplete coverage**
*Description:* Only a subset of affected paths, formats, or call sites is fixed; others remain broken. Typical signs include updating JSON but not XML, adjusting PRAGMA but not SELECT, fixing one code path while an equivalent exists elsewhere, or forgetting to update generated/runtime artifacts.

**C. Process hygiene and change discipline failures**
*Description:* The patch mixes unrelated edits (scope creep), alters tests to match a broken implementation, includes merge artifacts or duplicate code, or introduces syntax/typo/runtime errors (duplicate args, unreachable code). These complicate review and often obscure regressions.

**D. Contract/invariant violations or Abstraction/API misuse**
*Description:* Changes break explicit or implicit invariants or requirements. Examples include violating "single-column subquery," making non-atomic multi-step writes, changing multiplication order in non-abelian contexts, keeping multi-column projections inside IN subqueries, bypassing APIs or type contracts, or hardcoding internals. Also includes changing established behavior (defaults, tuple shapes, ordering, observable semantics) without justification or migration.

**E. Guard/safety-net removal or inversion**
*Description:* Removing or flipping checks, caches, or validation that protect correctness/security. Indicators include deleting `is_active` or `has_usable_password` checks, removing parent_link validation, dropping inverse/caching assignments, or disabling/inverting critical conditionals.

**F. Edge cases, normalization, and type/representation assumptions**
*Description:* Logic fails on uncommon values or conflates representations. Examples: treating `None` as the only "empty" (ignoring `"`), mishandling NaN/Inf or undefined semantics, missing lowercase exponent parsing, not rechecking length after mutation, confusing PATH vs PATH_INFO/script prefixes, or choosing wrappers/proxies that break expected type behavior. Includes overfitted regexes/parsers, missing named groups, unhandled array-indexed dispatch, naive SQL interpolation, missing escaping, off-by-one slices, or wrong encodings/BOM handling.

Descriptions used to categorise reviews -

**A. Functional correctness (logic, control flow, edge cases)**
*Description:* Ensure the fix implements the intended behavior with correct conditions, boundaries, ordering/precedence, and return values. Catch logic/sign errors, unreachable code, inverted conditions, and other correctness issues.

**B. Scope control and change isolation**
*Description:* Keep the patch tightly focused on the reported issue. Revert incidental edits, avoid broad refactors, and limit changes to the affected component or backend.

**C. API and data contract adherence**
*Description:* Preserve public/internal interfaces, data shapes, and semantics. Avoid breaking consumers, changing return types, or altering documented behavior without coordination.

**D. Design/architecture alignment and plumbing**
*Description:* Apply changes in the correct layer (e.g., model vs. view), respect separation of concerns, and route control flags/state through the call chain so policies are enforced where needed. Prefer non-breaking or backward-compatible design alternatives.

E. **Error and exception handling**

*Description:* Catch and handle expected failures at the correct layer; convert errors to appropriate no-ops or fallbacks. Avoid swallowing unexpected exceptions or leaking internal errors.

F. **Security and standards/protocol compliance**

*Description:* Use correct security checks (authz/authn, permission models), avoid unsafe operations (escaping, URL handling), and comply

## B  PROMPTS

**Review Generation**

```
1 You are an experienced software engineer tasked with
     reviewing code patches.
2 Below is a problem statement, a correct patch example, and a
      submitted patch which is likely incorrect or incomplete.
3 Please provide a detailed review of the submitted patch that
     identifies issues (e.g., missing context, incorrect
     modifications, or potential bugs) and specifies
     suggestions for improving the submitted patch so that it
     correctly solves the problem statement.
4 Avoid referencing the correct patch directly.
5
6 Problem Statement:
7 {{ problem_statement }}
8
9 Correct Patch Example:
10 {{ correct_patch_example }}
11
12 Submitted Patch (Bad Patch):
13 {{ bad_patch }}
14
15 Detailed Review:
```

**Bad Patch Genration**

```
1 You are given a production-ready source file below. Your
     task:
2 1. **Introduce one to two subtle, functional bugs** without
     adding any comments
3 2. **Do NOT break compilation** and **do not introduce any
     syntax or spelling errors** or make any code-style
     changes.
4 3. **Do NOT change any import statements**
5 4. Preserve formatting and comments; modify only the minimum
      lines needed to trigger a logical failure under certain
     inputs.
6 5. Return **only** the full modified file content, with no
     explanations or diff markers.
7
8 --- {path} original content START ---
9 {curr_text}
10 --- {path} original content END ---
```

**SWE-Agent Bug-fixing instructions**

```
1  <uploaded_files>
2  {{working_dir}}
3  </uploaded_files>
4  I've uploaded a python code repository in the directory {{
       working_dir}}. Consider the following PR description:
5
6  <pr_description>
7  {{problem_statement}}
8  </pr_description>
9
10 Can you help me implement the necessary changes to the
       repository so that the requirements specified in the <
       pr_description> are met?
11 I've already taken care of all changes to any of the test
       files described in the <pr_description>. This means you
       DON'T have to modify the testing logic or any of the
       tests in any way!
12 Your task is to make the minimal changes to non-tests files
       in the {{working_dir}} directory to ensure the <
       pr_description> is satisfied.
13 Follow these steps to resolve the issue:
14 1. As a first step, it might be a good idea to find and read
        code relevant to the <pr_description>
15 2. Create a script to reproduce the error and execute it
       with `python <filename.py>` using the bash tool, to
       confirm the error
16 3. Edit the sourcecode of the repo to resolve the issue
17 4. Rerun your reproduce script and confirm that the error is
        fixed!
18 5. Think about edgecases and make sure your fix handles them
        as well
19 Your thinking should be thorough and so it's fine if it's
       very long.
```

**SWE-Agent Test Generation instructions**

```
1  <uploaded_files>
2  {{working_dir}}
3  </uploaded_files>
4  I've uploaded a python code repository in the directory {{
     working_dir}}. Consider the following problem description
     :
5
6  <problem_description>
7  {{problem_statement}}
8  </problem_description>
9
10 Can you help me implement a test that successfully
     reproduces the problem specified in the <
     problem_description>?
11 The test must be created in the repository's existing test
     suite and should be runable with the repository's testing
      infrastructure / tooling (e.g. pytest).
12 Do not make any changes to the non-test code in the
     repository since we only need to create a reproduction
     test.
13 Follow these steps to resolve the issue:
14 1. As a first step, it might be a good idea to find and read
      code relevant to the <problem_description>
15 2. Create a script to reproduce the error and execute it
     with `python <filename.py>` using the bash tool, to
     confirm the error
16 3. Edit the the testing suite of the repo to implement a
     test based on this reproduction script which can be run
     using the repository's testing infrastructure / tooling (
     e.g. pytest)
17 4. Ensure this test runs and successfully reproduces the
     problem!
18 5. Remove the reproduction script and only keep changes to
     the test suite that reproduce the problem.
19 Your thinking should be thorough and so it's fine if it's
     very long.
```

**SWE-Agent Style-Fix instructions**

```
1  You have recently generated a patch to resolve an issue
       within this repository.
2  Pylint has been run on the modified files and has produced
       the following feedback:
3
4  {{problem_statement}}
5
6  Your task is to:
7  1. Analyze the Pylint violations provided in the problem
       statement
8  2. Understand the specific rules that were violated (e.g.,
       naming conventions, unused imports, complexity issues)
9  3. Apply fixes that resolve these errors while maintaining
       code functionality
10 4. Ensure your changes follow Python best practices and
       improve code readability
11 5. Test that your fixes don't introduce new Pylint
       violations
12 6. Do not introduce any new files to fix the style errors
13
14 Common Pylint violations you may encounter:
15 - Naming and style issues (invalid-name, missing-docstring,
       line-too-long)
16 - Import issues (unused-import, wrong-import-order,
       reimported)
17 - Error-prone patterns (undefined-variable, no-member,
       unsubscriptable-object)
18 - Code design issues (too-many-arguments, too-many-locals,
       too-many-branches)
19 - Best practice and maintainability issues (fixme, unused-
       argument, broad-except)
20
21 Please resolve the Pylint feedback to the best of your
       ability, while preserving the functionality of the code.
22 Focus on the most critical violations first and ensure your
       fixes improve overall code quality and maintainability.
```

**SWE-Agent Review-Fix instructions**

```
1  <uploaded_files>
2  {{working_dir}}
3  </uploaded_files>
4  I've uploaded a code repository in the directory {{
       working_dir}}. {{problem_statement}}
5
6  Can you help me implement the necessary changes to the
       repository so that the requirements specified in the <
       pr_description> are met?
7  I've already taken care of all changes to any of the test
       files described in the <pr_description>. This means you
       DON'T have to modify the testing logic or any of the
       tests in any way!
8  Your task is to make the minimal changes to non-tests files
       in the {{working_dir}} directory to ensure the <
       pr_description> is satisfied.
9  Follow these steps to resolve the issue:
10 1. As a first step, it might be a good idea to find and read
        code relevant to the <pr_description>
11 2. Create a script to reproduce the error and execute it to
       confirm the error
12 3. Edit the sourcecode of the repo to resolve the issue
13 4. Rerun your reproduce script and confirm that the error is
        fixed!
14 5. Think about edgecases and make sure your fix handles them
        as well
15 Your thinking should be thorough and so it's fine if it's
       very long.
```

# C  STYLE REVIEW

## C.1  STYLE REVIEW SCORE ANALYSIS

Here, we provide additional information on how LLMs perform on Style Fixing tasks independent of functionality. The metrics and their formulas are:

$$\text{Fix Rate} = \frac{\text{number of resolved original errors}}{\text{number of original errors}}$$

$$\text{Error Ratio} = \frac{\text{number of original errors} - \text{number of resolved original errors} + \text{number of new errors created}}{\text{number of original errors}}$$

$$\text{Overall Fix Rate} = \frac{\text{number of resolved original errors}}{\text{number of original errors} + \text{number of new errors created}}$$

Table 4: Style Review Score Analysis

| Language | Experimental Setting | Fix Rate | Error Ratio | Overall Fix Rate | Score |
|---|---|---|---|---|---|
| Python | SWE-Agent + Gemini-2.5-Flash | 96.2% | 0.377 | 80.1% | 72.2% -> $57.0\% \pm 6.9$ |
| | SWE-Agent + DeepSeek-V3.1 | 91.5% | 0.299 | 79.5% | 73.4% -> $54.0\% \pm 7.2$ |
| | SWE-Agent + GPT-5-mini | 65.3% | 0.457 | 59.6% | 56.3% -> $45.9\% \pm 7.7$ |
| | SWE-Agent + Qwen3-32B | 30.5% | 0.891 | 25.5% | 22.7% -> $19.5\% \pm 6.2$ |
| | Aider + Gemini-2.5-Flash | 85.7% | 0.482 | 69.3% | 60.3% -> $48.6\% \pm 7.0$ |
| C++ | SWE-Agent + Gemini-2.5-Flash | 75.9% | 2.49 | 48.7% | |
| | SWE-Agent + DeepSeek-V3.1 | 68.0% | 2.46 | 41.4% | |
| | SWE-Agent + GPT-5-mini | 47.3% | 2.61 | 28.6% | |
| | SWE-Agent + Qwen3-32B | 35.3% | 2.87 | 18.1% | |
| | Aider + Gemini-2.5-Flash | 25.1% | 2.82 | 15.3% | –% |
| Java | SWE-Agent + Gemini-2.5-Flash | 80.9% | 5.17 | 40.4% | |
| | SWE-Agent + DeepSeek-V3.1 | 77.9% | 5.46 | 36.8% | |
| | SWE-Agent + GPT-5-mini | 64.1% | 5.38 | 35.2% | |
| | SWE-Agent + Qwen3-32B | 66.0% | 5.31 | 34.5% | |
| | Aider + Gemini-2.5-Flash | 81.2% | 5.55 | 37.6% | –% |

## C.2  RULESETS USED FOR STYLE REVIEW

Table 5: List of Python Style Errors.

| | | |
|---|---|---|
| protected-access | redefined-outer-name | unused-argument |
| attribute-defined-outside-init | abstract-method | fixme |
| redefined-builtin | invalid-str-returned | unused-variable |
| anomalous-backslash-in-string | unnecessary-pass | broad-exception-caught |
| raise-missing-from | unbalanced-tuple-unpacking | arguments-differ |
| unused-import | reimported | assigning-non-slot |
| unnecessary-lambda | undefined-variable | pointless-statement |
| logging-fstring-interpolation | missing-timeout | unsubscriptable-object |
| logging-not-lazy | pointless-string-statement | not-callable |
| unspecified-encoding | dangerous-default-value | invalid-field-call |
| possibly-used-before-assignment | arguments-renamed | eval-used |
| no-self-argument | unexpected-keyword-arg | bare-except |
| too-many-function-args | no-value-for-parameter | expression-not-assigned |
| cell-var-from-loop | comparison-with-callable | super-init-not-called |
| undefined-loop-variable | used-before-assignment | global-variable-not-assigned |
| abstract-class-instantiated | access-member-before-definition | bad-staticmethod-argument |
| deprecated-class | function-redefined | implicit-str-concat |
| not-context-manager | signature-differs | super-without-brackets |
| invalid-unary-operand-type | broad-exception-raised | arguments-out-of-order |
| assert-on-string-literal | bad-indentation | global-statement |
| global-variable-undefined | import-self | invalid-getnewargs-ex-returned |
| invalid-metaclass | invalid-repr-returned | invalid-sequence-index |
| isinstance-second-argument-not-valid-type | keyword-arg-before-vararg | misplaced-bare-raise |
| missing-kwoa | non-parent-init-called | possibly-unused-variable |
| raising-non-exception | redundant-u-string-prefix | redundant-unittest-assert |
| subprocess-run-check | unnecessary-ellipsis | unused-private-member |
| wildcard-import | astroid-error | syntax-error |
| useless-parent-delegation | bad-super-call | method-hidden |
| not-an-iterable | too-few-format-args | assignment-from-no-return |
| assignment-from-none | bad-chained-comparison | bad-str-strip-call |
| bad-string-format-type | bad-thread-instantiation | bidirectional-unicode |
| contextmanager-generator-missing-cleanup | deprecated-argument | deprecated-method |
| deprecated-module | dict-iter-missing-items | duplicate-except |
| duplicate-key | duplicate-string-formatting-argument | duplicate-value |
| exec-used | f-string-without-interpolation | format-string-without-interpolation |
| inherit-non-class | invalid-bool-returned | invalid-length-returned |
| invalid-overridden-method | logging-format-interpolation | logging-too-many-args |
| lost-exception | method-cache-max-size-none | modified-iterating-list |
| nested-min-max | no-method-argument | non-iterator-returned |
| notimplemented-raised | pointless-exception-statement | positional-only-arguments-expected |
| raising-bad-type | raising-format-tuple | redeclared-assigned-name |
| redundant-keyword-arg | return-in-finally | return-in-init |
| self-assigning-variable | self-cls-assignment | shadowed-import |
| try-except-raise | unbalanced-dict-unpacking | undefined-all-variable |
| unexpected-special-method-signature | unnecessary-semicolon | unpacking-non-sequence |
| unreachable | unsupported-assignment-operation | unsupported-delete-operation |
| unsupported-membership-test | unused-format-string-argument | unused-wildcard-import |
| used-prior-global-declaration | useless-else-on-loop | using-constant-test |

Table 6: List of Java Style Errors

| | | |
|---|---|---|
| AtLeastOneConstructor | AvoidDuplicateLiterals | CommentDefaultAccessModifier |
| FieldNamingConventions | LawOfDemeter | LocalVariableCouldBeFinal |
| MethodArgumentCouldBeFinal | OnlyOneReturn | ShortClassName |
| UnnecessaryImport | UseUtilityClass | AvoidCatchingGenericException |
| AvoidDeeplyNestedIfStmts | AvoidLiteralsInIfCondition | ClassWithOnlyPrivateConstructorsShouldBeFinal |
| JUnitTestContainsTooManyAsserts | JUnitTestsShouldIncludeAssert | LinguisticNaming |
| SystemPrintln | TestClassWithoutTestCases | AvoidAccessibilityAlteration |
| AvoidCatchingThrowable | CallSuperInConstructor | CognitiveComplexity |
| ImmutableField | LooseCoupling | ShortMethodName |
| SignatureDeclareThrowsException | TooManyStaticImports | UseDiamondOperator |
| UseUnderscoresInNumericLiterals | UselessParentheses | AssignmentInOperand |
| AvoidFieldNameMatchingMethodName | AvoidReassigningParameters | AvoidThrowingRawExceptionTypes |
| CollapsibleIfStatements | ConfusingTernary | CouplingBetweenObjects |
| CyclomaticComplexity | DataClass | ExceptionAsFlowControl |
| ExcessivePublicCount | GodClass | LiteralsFirstInComparisons |
| MethodNamingConventions | MutableStaticState | NPathComplexity |
| NcssCount | NullAssignment | PreserveStackTrace |
| SimplifyBooleanReturns | TooManyFields | UnnecessaryBoxing |
| UnnecessaryConstructor | UnusedFormalParameter | UseProperClassLoader |
| AbstractClassWithoutAbstractMethod | ArrayIsStoredDirectly | AvoidBranchingStatementAsLastInLoop |
| ClassNamingConventions | CloseResource | CompareObjectsWithEquals |
| EmptyCatchBlock | ExcessiveImports | FieldDeclarationsShouldBeAtStartOfClass |
| ForLoopCanBeForeach | JUnit4TestShouldUseTestAnnotation | LocalVariableNamingConventions |
| OneDeclarationPerLine | ReturnEmptyCollectionRatherThanNull | UnnecessaryFullyQualifiedName |
| UnnecessaryReturn | UnnecessarySemicolon | UnusedAssignment |
| UseTryWithResources | UseVarargs | AvoidFieldNameMatchingTypeName |
| AvoidReassigningLoopVariables | AvoidUncheckedExceptionsInSignatures | ControlStatementBraces |
| EmptyControlStatement | GenericsNaming | GuardLogStatement |
| MethodReturnsInternalArray | PrematureDeclaration | SwitchStmtsShouldHaveDefault |
| UnnecessaryCast | UnnecessaryModifier | UnusedPrivateMethod |
| UseLocaleWithCaseConversions | UseShortArrayInitializer | AvoidThrowingNullPointerException |
| BooleanGetMethodName | ConstantsInInterface | ConstructorCallsOverridableMethod |
| ExcessiveParameterList | FinalFieldCouldBeStatic | ForLoopVariableCount |
| JUnitUseExpected | MissingSerialVersionUID | NonStaticInitializer |
| OverrideBothEqualsAndHashcode | UnnecessaryAnnotationValueElement | UnnecessaryLocalBeforeReturn |
| UnusedLocalVariable | UseCollectionIsEmpty | UseEqualsToCompareStrings |
| UseStandardCharsets | AbstractClassWithoutAnyMethod | AvoidCatchingNPE |
| AvoidProtectedFieldInFinalClass | AvoidProtectedMethodInFinalClassNotExtending | AvoidUsingHardCodedIP |
| DoubleBraceInitialization | EmptyMethodInAbstractClassShouldBeAbstract | EqualsNull |
| FormalParameterNamingConventions | ImplicitSwitchFallThrough | JUnit5TestShouldBePackagePrivate |
| MissingStaticMethodInNonInstantiatableClass | ReplaceVectorWithList | SimpleDateFormatNeedsLocale |
| SimplifiedTernary | SwitchDensity | AvoidDecimalLiteralsInBigDecimalConstructor |
| AvoidDollarSigns | AvoidInstanceofChecksInCatchClause | AvoidPrintStackTrace |
| AvoidRethrowingException | AvoidStringBufferField | DoNotCallGarbageCollectionExplicitly |
| DontImportSun | FinalParameterInAbstractMethod | IdenticalCatchBranches |
| MissingOverride | NonSerializableClass | PrimitiveWrapperInstantiation |
| SuspiciousEqualsMethodName | UnusedPrivateField | AvoidThrowingNewInstanceOfSameException |
| DefaultLabelNotLastInSwitchStmt | DetachedTestCase | DoNotExtendJavaLangThrowable |
| DoNotTerminateVM | ForLoopShouldBeWhileLoop | InstantiationToGetClass |
| JUnit4SuitesShouldUseSuiteAnnotation | JumbledIncrementer | LogicInversion |
| ProperCloneImplementation | ReplaceHashtableWithMap | SimplifyBooleanExpressions |
| SimplifyConditional | SingletonClassReturningNewInstance | SingularField |
| UseObjectForClearerAPI | AssignmentToNonFinalStatic | AvoidMessageDigestField |
| AvoidMultipleUnaryOperators | AvoidUsingOctalValues | CheckSkipResult |
| ClassCastExceptionWithToArray | CloneMethodMustBePublic | CloneMethodMustImplementCloneable |
| CloneMethodReturnTypeMustMatchClassName | DoNotExtendJavaLangError | DoNotHardCodeSDCard |
| DoNotThrowExceptionInFinally | DontUseFloatTypeForLoopIndices | FinalizeDoesNotCallSuperFinalize |
| InvalidLogMessageFormat | NoPackage | PackageCase |
| SingleMethodSingleton | UnconditionalIfStatement | UnnecessaryCaseChange |
| UnusedNullCheckInEquals | UseExplicitTypes | UselessOperationOnImmutable |
| UselessOverridingMethod | UselessQualifiedThis | WhileLoopWithLiteralBoolean |

Table 7: List of CPP Style Errors

| | |
|---|---|
| misc-include-cleaner | misc-use-anonymous-namespace |
| cppcoreguidelines-avoid-magic-numbers | cppcoreguidelines-avoid-do-while |
| misc-const-correctness | cppcoreguidelines-rvalue-reference-param-not-moved |
| misc-non-private-member-variables-in-classes | bugprone-easily-swappable-parameters |
| cppcoreguidelines-pro-bounds-pointer-arithmetic | cppcoreguidelines-avoid-c-arrays |
| cppcoreguidelines-avoid-non-const-global-variables | cppcoreguidelines-pro-bounds-array-to-pointer-decay |
| cppcoreguidelines-owning-memory | cppcoreguidelines-init-variables |
| cppcoreguidelines-macro-usage | cppcoreguidelines-special-member-functions |
| cppcoreguidelines-pro-type-member-init | cppcoreguidelines-pro-type-static-cast-downcast |
| misc-no-recursion | performance-enum-size |
| bugprone-narrowing-conversions | cppcoreguidelines-narrowing-conversions |
| cppcoreguidelines-pro-type-reinterpret-cast | cppcoreguidelines-pro-type-union-access |
| cppcoreguidelines-use-default-member-init | cppcoreguidelines-pro-bounds-constant-array-index |
| bugprone-implicit-widening-of-multiplication-result | bugprone-macro-repeated-side-effects |
| bugprone-suspicious-include | clang-analyzer-optin.core.EnumCastOutOfRange |
| cppcoreguidelines-avoid-const-or-ref-data-members | cppcoreguidelines-explicit-virtual-functions |
| cppcoreguidelines-pro-type-vararg | portability-simd-intrinsics |

## C.3   STYLE REVIEW ERROR ANALYSIS

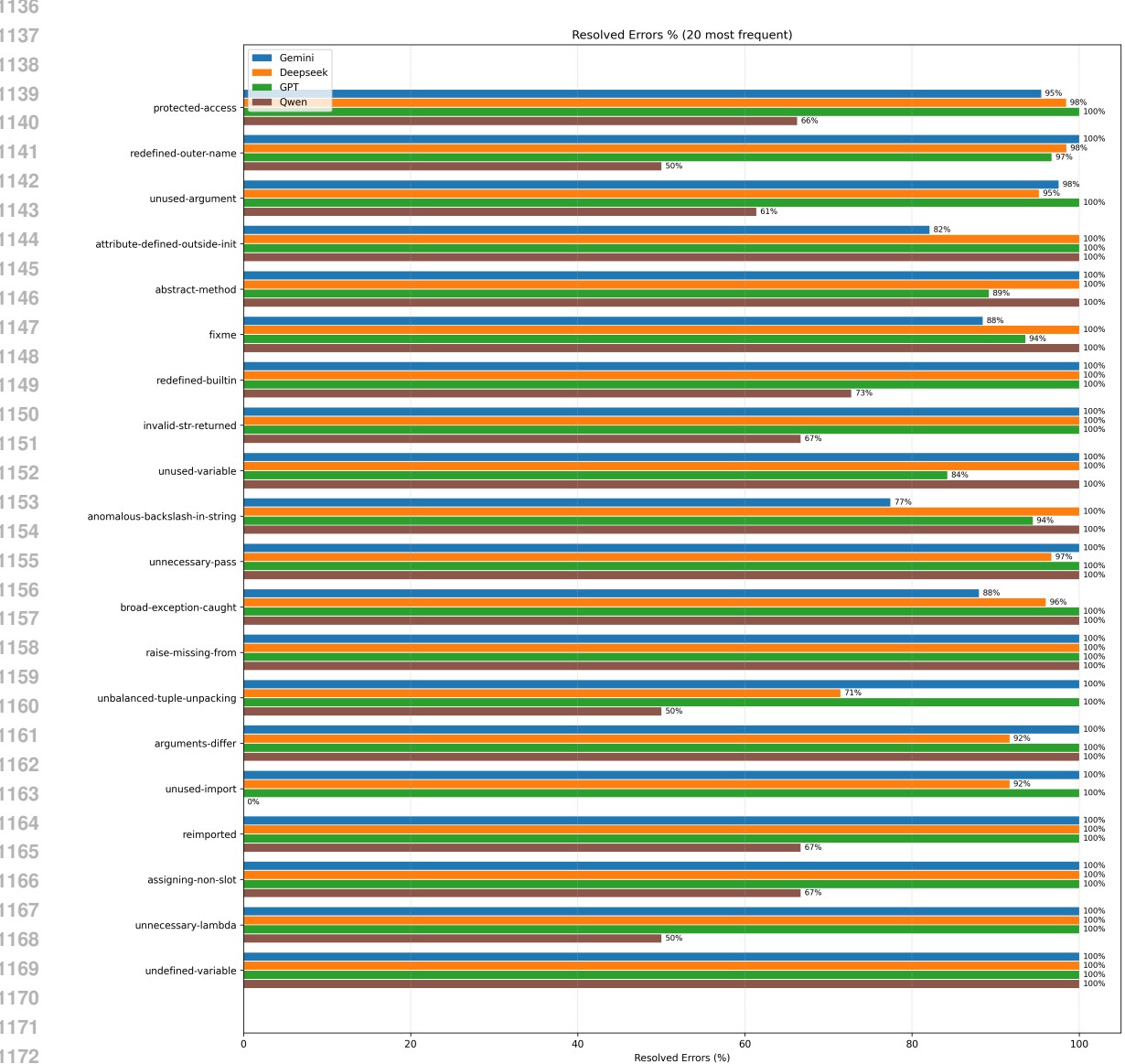

Figure 8: Resolve rates for the 20 most frequent style errors in Python.

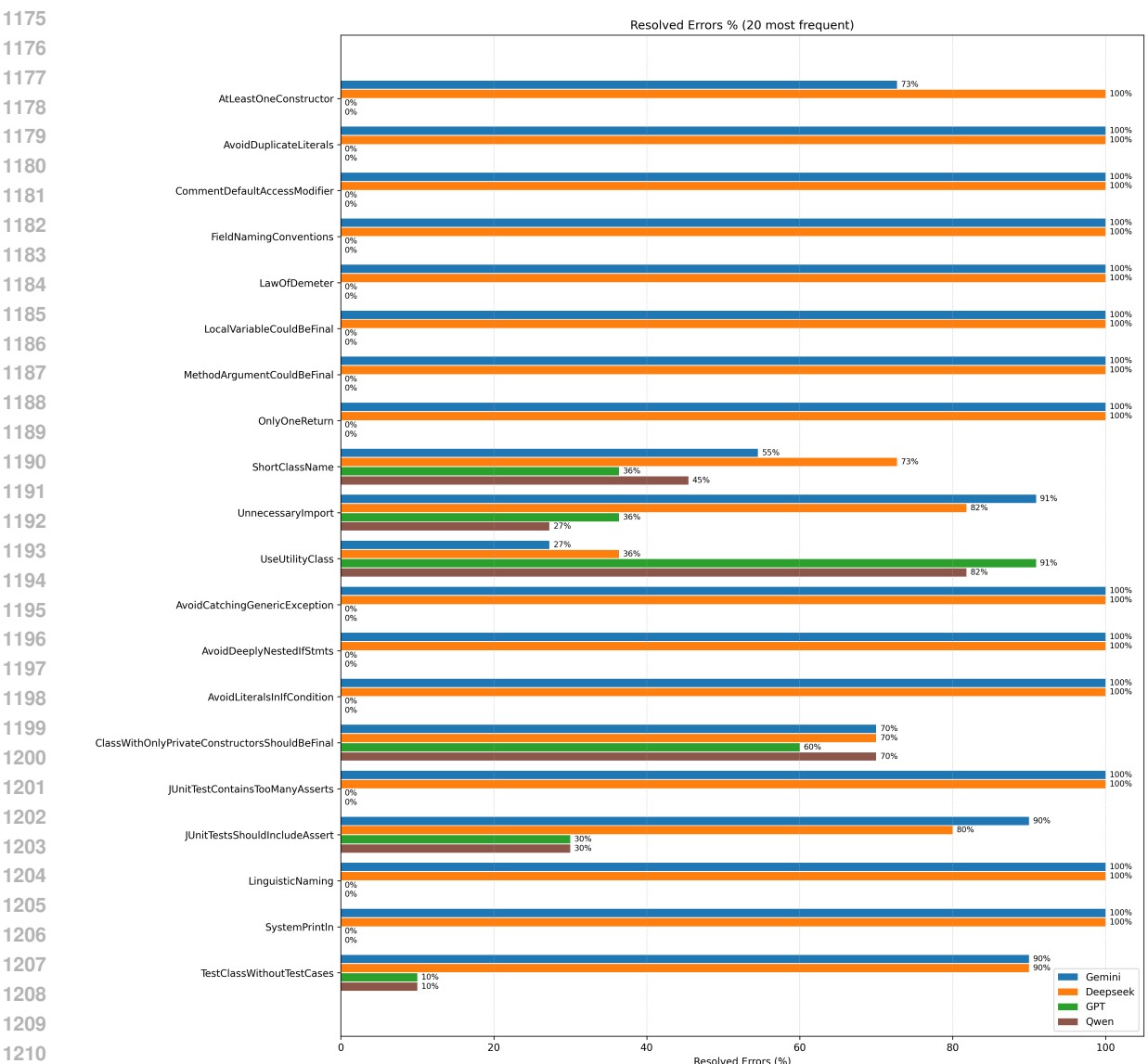

Figure 9: Resolve rates for the 20 most frequent style errors in Java.

As shown in 9, agents reliably fix local, syntactic style issues but diverge on semantic or cross-file refactorings. For example, many rules — AvoidDuplicateLiterals, CommentDefaultAccessModifier, FieldNamingConventions, LawOfDemeter, LocalVariableCouldBeFinal, MethodArgumentCould-BeFinal, OnlyOneReturn, AvoidCatchingGenericException, AvoidDeeplyNestedIfStmts, — are resolved at 100% by all three agents, which indicates these errors stem from local, pattern-detectable oversights (leftover literals, missing modifiers, simple nesting or println usages) and can be corrected by single-file, syntactic edits or well-scoped templates. By contrast, errors that require either boiler-plate insertion or light architectural judgement show agent differences: AtLeastOneConstructor is

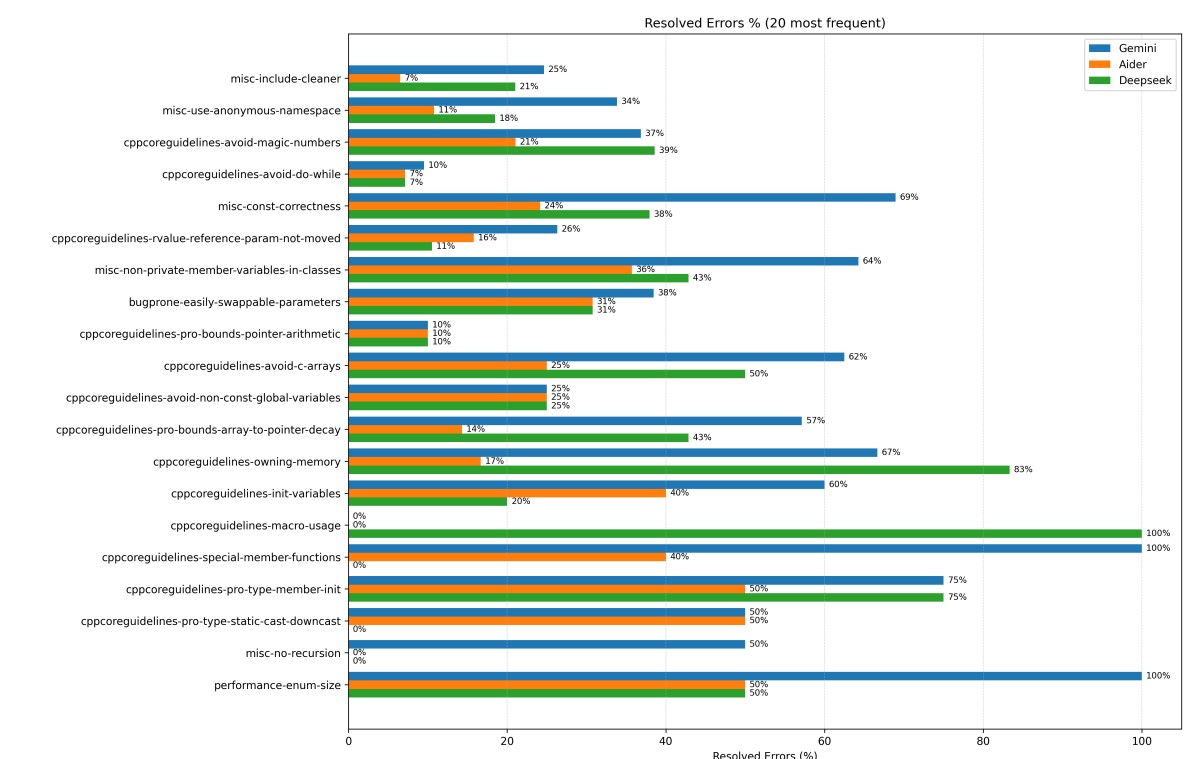

Figure 10: Resolve rates for the 20 most frequent style errors in CPP.

resolved by Gemini 73%, Aider 82% and Deepseek 100% (Deepseek appears strongest at inserting appropriate constructors, suggesting it better synthesizes class skeletons), UnnecessarilyImport is handled best by Gemini (91% vs ~82%), which implies Gemini is particularly effective at mechanical cleanup (removing IDE-leftover imports), while UseUtilityClass is the hardest (Gemini 27%, Aider 64%, Deepseek 36%) — converting a class to a utility requires semantic understanding (that methods are stateless/should be static and constructors removed), project-wide implications and non-trivial refactoring heuristics, so performance drops. ShortClassName (Gemini 55%, Aider/Deepseek ~73%) and ClassWithOnlyPrivateConstructorsShouldBeFinal (Gemini/Deepseek ~70%, Aider 80%) similarly reflect refactor/semantic sensitivity: these errors occur because of design choices (poor naming, classes meant as singletons/factories) and hence need broader context or safer rename patterns to fix without breaking references. Finally, test-related fixes (JUnitTestsShouldIncludeAssert: Gemini 90% vs others ~80%; TestClassWithoutTestCases ~90% all) show that adding assertions or test content is approachable but benefits from an agent's ability to infer test intent. Thus, if the fix is a local, syntactic removal or modifier change (the common result of IDE habits or quick edits) all agents excel; when the fix requires synthesis of new boilerplate or a design-level judgement (constructors, utility conversion, safe renames), performance diverges and the better agent is the one that more reliably infers program intent and can safely make cross-site edits — exactly the kinds of capabilities we should prioritize next in automated style repair.

In 10, a clear partitioning of agent capability emerges. Gemini attains the highest resolution rates on checks that are syntactically local and mechanically canonical—notably misc-const-correctness,

misc-non-private-member-variables-in-classes and cppcoreguidelines-avoid-c-arrays, indicating it reliably performs small, deterministic AST-level edits where the root cause is programmer oversight or legacy C idioms. Deepseek dominates categories tied to legacy manual-memory and preprocessor practices—most prominently cppcoreguidelines-owning-memory and cppcoreguidelines-macro-usage —which directly implies it is better at recognizing and applying idiomatic modernization or conservative rewrites in codebases where errors stem from explicit new/delete patterns and heavy macro usage. Aider occupies an intermediate regime with moderate resolve rates on initialization and type-related checks (cppcoreguidelines-init-variables, cppcoreguidelines-pro-type-member-init), suggesting a propensity for lower-risk, surface-level repairs rather than broad structural refactors. Across agents, the highest absolute resolve rates correspond to mechanically fixable, single-rewrite problems (local syntactic omissions or replace-with-standard-container transformations), whereas checks that require understanding programmer intent, cross-cutting design choices, or semantic refactoring exhibit lower and more variable resolution; this pattern directly traces to the origin of each error class—simple oversight or legacy idiom versus deep semantic or intentional ambiguity—and implies that improving automated style repair requires either stronger intent inference (tests, specifications) or broader, transformation-aware training focused on non-local semantic refactors. This can be surmised from the fact that errors such as misc-const-correctness, misc-non-private-member-variables-in-classes, cppcoreguidelines-avoid-c-arrays, cppcoreguidelines-pro-bounds-array-to-pointer-decay and cppcoreguidelines-pro-type-member-init performance-enum-size exhibit consistently high resolve rates (with Gemini leading on several), whereas other checks show moderate to low and often heterogeneous performance across agents. These high-rate rows correspond to local, syntactic, single-step transformations - adding `const`, restricting member visibility, or replacing raw C arrays with standard containers - whose root causes are programmer oversight or legacy C idioms and therefore admit deterministic AST-level repairs. By contrast, rows with low or mixed resolution reflect checks that demand cross-cutting reasoning about ownership, lifetime, or design intent; their failure modes in the plot indicate semantic ambiguity rather than simple syntactic omission. Consequently, the visual evidence supports the interpretation that automated style repair succeeds where a canonical, local rewrite exists and degrades where fixes require intent inference or non-local semantic refactoring.

Further it can be seen though, the agents have a higher resolve rate for Java style errors, they are also prone to introduce more number of additional errors as compared to resolving CPP style errors.

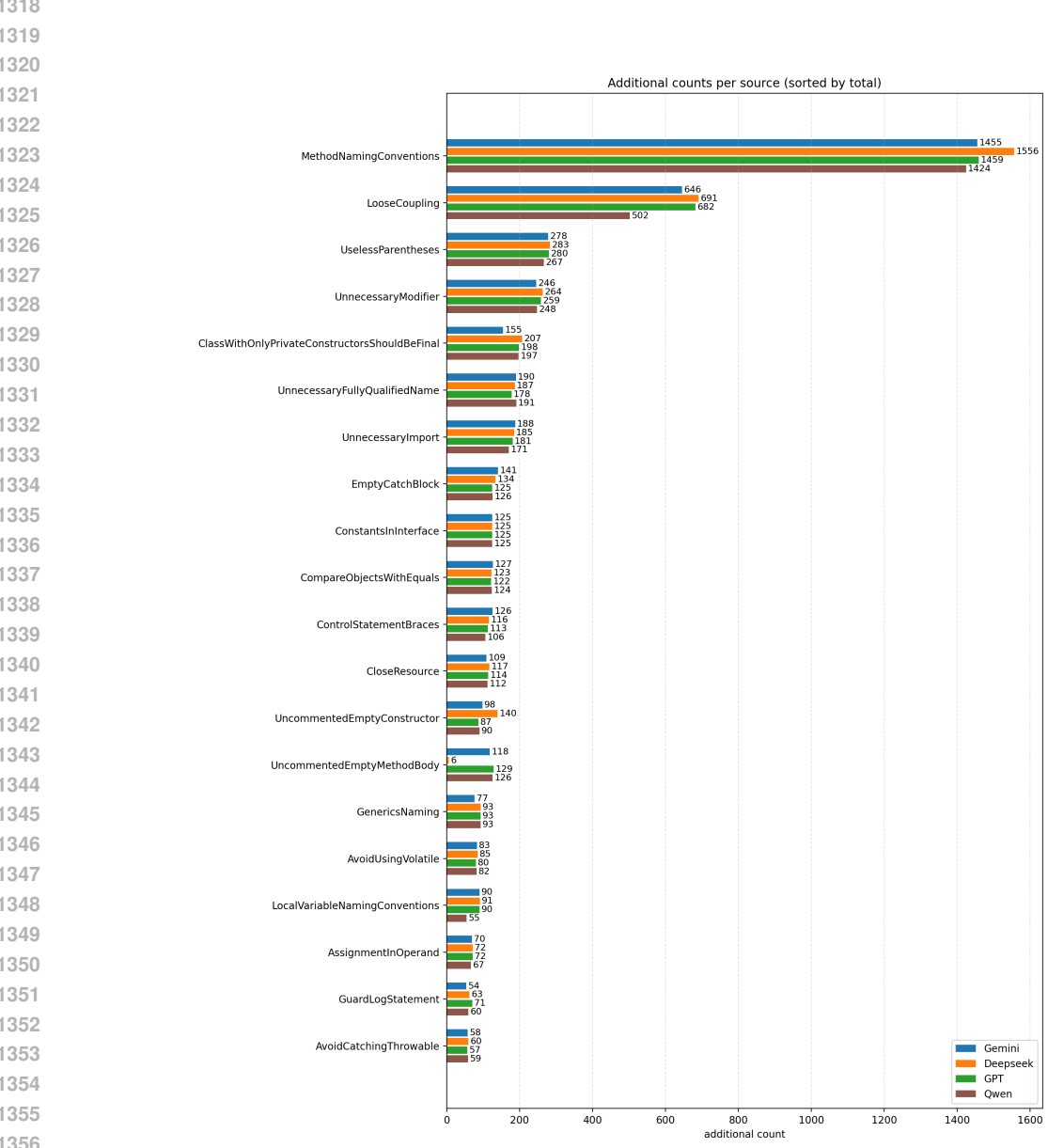

Figure 11: Counts for the 20 most frequent additional errors in Java.

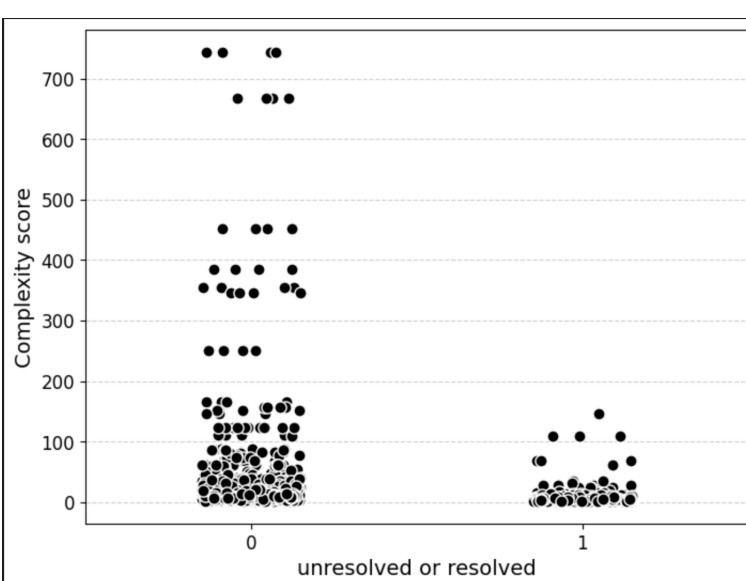

Figure 12: Distribution of Complexity Score between Resolved and Unresolved Instances

## D    PATCH COMPLEXITY ANALYSIS

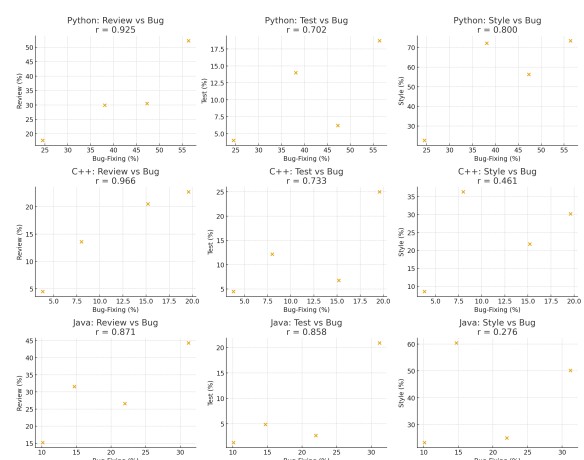

Figure 13: Bugfixing performance ploted together with performance on other tasks seperately for each language.

# E    CORRELATION ANALYSIS ACROSS TASKS

Table 8: Per-language Pearson correlation between Bug-Fixing and other tasks

| Language | Review vs Bug | Test vs Bug | Style vs Bug |
|----------|---------------|-------------|--------------|
| Python   | 0.925         | 0.702       | 0.800        |
| C++      | 0.966         | 0.733       | 0.461        |
| Java     | 0.871         | 0.858       | 0.276        |
| **Average** | **0.921**  | **0.764**   | **0.512**    |

# F  PATCH GENERATION RATE

Table 9: SWE-Agent Patch Generate Rate across models and tasks

| Language | Model | Bug-Fixing | Test-Generation | Review-Response | Style-Fixing |
|---|---|---|---|---|---|
| Python | Gemini-2.5-Flash | 93.8% | – | 92.7% | 91.7% |
| | DeepSeek-V3.1 | 96.3% | 94.8% | 95.1% | 93.8% |
| | GPT-5-mini | 76.2% | 64.8% | 61.0% | 69.3% |
| | Qwen3-32B | 79.1% | 90.8% | 78.0% | 35.7% |
| C++ | Gemini-2.5-Flash | 98.2% | 97.7% | 77.3% | 80.3% |
| | DeepSeek-V3.1 | 96.4% | 75.0% | 97.7% | 85.7% |
| | GPT-5-mini | 62.5% | 54.5% | 56.8% | 48.3% |
| | Qwen3-32B | 70.5% | 97.7% | 88.6% | 47.6% |
| Java | Gemini-2.5-Flash | 99.1% | 79.2% | 93.7% | 84.7% |
| | DeepSeek-V3.1 | 93.6% | 87.0% | 91.1% | 91.1% |
| | GPT-5-mini | 45.9% | 45.5% | 51.9% | 50.0% |
| | Qwen3-32B | 75.2% | 90.9% | 77.2% | 44.4% |

Table 10: Bugfixing - Avg. Complexity Score.

| Model | Language | Gold Avg Complexity | Model Avg Complexity | Resolved Gold Avg Complexity | Resolved Model Avg Complexity | Unresolved Gold Avg Complexity | Unresolved Model Avg Complexity |
|---|---|---|---|---|---|---|---|
| Gemini 2.5 Flash | Python | 7.07 | 299.28 | 5.35 | 5.28 | 8.13 | 484.67 |
| | Java | 19.24 | 9.75 | 6.69 | 12.32 | 19.67 | 19.24 |
| | C++ | 47.55 | 195.1 | 8.07 | 6.28 | 38.26 | 252.31 |
| Deepseek v3.1 | Python | 7.07 | 12.08 | 5.22 | 5.35 | 9.46 | 21.60 |
| | Java | 19.24 | 12.91 | 6.47 | 7.07 | 26.49 | 15.84 |
| | C++ | 47.55 | 104.63 | 9.21 | 32.13 | 54.29 | 123.18 |
| GPT-5-mini | Python | 7.07 | 165.56 | 4.30 | 4.05 | 9.55 | 390.18 |
| | Java | 19.24 | 983.12 | 6.51 | 4.24 | 21.95 | 1186.70 |
| | C++ | 47.55 | 603.39 | 18.48 | 90.91 | 43.92 | 767.78 |
| qwen3-32b | Python | 7.07 | 464.93 | 5.77 | 4.32 | 7.49 | 642.09 |
| | Java | 19.24 | 4.76 | 5.26 | 2.7 | 24.28 | 5.08 |
| | C++ | 47.55 | 140.96 | 5.00 | 4.75 | 46.37 | 148.22 |

Table 11: Review-Response - Avg. Complexity Score.

| Model | Language | Gold Avg Complexity | Model Avg Complexity | Resolved Gold Avg Complexity | Resolved Model Avg Complexity | Unresolved Gold Avg Complexity | Unresolved Model Avg Complexity |
|---|---|---|---|---|---|---|---|
| Gemini 2.5 flash | Python | 7.07 | 1635.47 | 3.69 | 9.58 | 7.93 | 2408.95 |
| | Java | 19.24 | 9.95 | 6.49 | 6.86 | 17.25 | 11.53 |
| | C++ | 47.55 | 128.33 | 5.98 | 4.85 | 41.85 | 154.79 |
| Deepseek v3.1 | Python | 7.07 | 10.71 | 4.36 | 6.24 | 9.23 | 16.26 |
| | Java | 19.24 | 9.75 | 6.69 | 7.04 | 19.67 | 12.32 |
| | C++ | 47.55 | 195.10 | 8.07 | 6.28 | 38.26 | 252.31 |
| GPT-5-mini | Python | 7.07 | 289.22 | 4.96 | 13.80 | 7.40 | 543.45 |
| | Java | 19.24 | 6.26 | 6.91 | 5.58 | 15.32 | 6.99 |
| | C++ | 47.55 | 955.24 | 9.01 | 5.84 | 12.39 | 1489.28 |
| qwen3-32b | Python | 7.07 | 519.86 | 3.15 | 3.26 | 7.41 | 639.08 |
| | Java | 19.24 | 3.96 | 5.85 | 2.83 | 16.83 | 4.31 |
| | C++ | 47.55 | 248.65 | 2.25 | 2.4 | 14.55 | 261.96 |

