# OpenReview forum: "OmniCode: A Benchmark for Evaluating Software Development Agents"
_ICLR.cc/2026/Conference — ICLR 2026 Conference Withdrawn Submission_

### Official Review · Reviewer_8Gg7 · 2025-10-30

**Soundness:** 2
**Presentation:** 2
**Contribution:** 3
**Rating:** 2
**Confidence:** 4

**Summary:**

This paper introduces OmniCode, a benchmark for evaluating LLM-powered software development agents across four task categories: bug fixing, test generation, code review response, and style fixing. The benchmark comprises 1,794 tasks spanning Python, Java, and C++, derived from 494 base instances. The authors evaluate popular agent scaffolding frameworks (SWE-Agent and Aider) and find significant performance gaps, particularly in test generation and C++ tasks.

The idea of extending task types beyond bug-fixing or feature development is timely and addresses a real gap in existing benchmarks. The work builds on (Multi-)SWE-Bench and introduces methods to synthetically generate multiple task types from already collected instances. However, the work has several limitations that weaken its contribution: the pipeline's manual curation process limits scalability to other languages; the experimental evaluation is restricted to a single model family (Gemini); validation of synthetically generated components (bad patches, code reviews) is insufficient; statistical analysis of results is missing; and critical reproducibility information (prompts, code, containers) is not provided in the submission.

**Strengths:**

1. Adding code review response, test generation, and style-fixing alongside bug repair usefully widens evaluation beyond SWE-Bench-like tasks and represents a step toward more comprehensive software engineering evaluation.
2. The multi-bad-patch protocol for test evaluation is a meaningful design choice that ensures generated tests are non-trivial.
3. Using not only SWE-Bench but also Multi-SWE-Bench enables support for popular languages beyond Python.
4. The approach of bootstrapping multiple task types from base instances is compelling and enables large-scale automation.
5. The overall structure is easy to follow.

**Weaknesses:**

1. The paper lacks ablations and additional analyses to validate the synthetic data generation pipeline. Since synthetic data is a major component, it may be sensitive to different prompts, introduce leakage, or create unsolvable tasks. Specifically:
  - The paper states that code reviews are "informative but do not give away the complete solutions" but provides no rigorous evaluation of review quality, realism, or usefulness.
  - No validation is provided for the bad patches, are they realistic failure modes that strong models would produce?
2. The primary evaluation is quite limited. While SWE-Agent and Aider can be considered representative examples of current SWE scaffolding approaches, evaluating only with Gemini 2.5 Flash is insufficient. Comparison with leading open-source models (Qwen3-Coder/GLM-4.x/Kimi-K2) and ideally frontier models (gpt/sonnet) would provide a more comprehensive understanding of how modern LLMs perform on the proposed benchmark.
3. SWE agents' performance can vary significantly across runs, especially on small subsets like the 44 C++ instances for test generation. Reporting confidence intervals or standard errors would provide stronger statistical evidence for the performance claims.
4. The conclusions lack detail. For example, the paper states "we observe that it struggles at C++ tasks as well as Test-Generation across languages" but provides minimal investigation into underlying causes. What specific patterns emerge in test generation failures? What types of C++ bugs are most challenging?
5. The observation that reviews help for Java/C++ but hurt for Python is interesting but remains unexplained. The speculation about "distraction" lacks empirical support and appears speculative.
6. While the appendix is referenced, critical details for reproducibility are missing from the provided excerpt, so it seems that appendix itself is missing.
  - The actual prompts used for each task type are not included
  - The bad patch generation prompt is referenced but not shown
7. [Minor] The paper contains a series of typos, e.g., guage -> gauge, incomplete sentence in Sec. 5

**Questions:**

1. What specific patterns emerge in test generation failures? Are agents failing to understand the bug, unable to construct proper test syntax, or missing edge cases?
2. The formula (ΔFiles + Hunks + AddedLines + RemovedLines)/10 lacks theoretical or empirical justification. Could you explain how this formula was derived? Different components seem to have vastly different scales (e.g., ΔFiles typically ranges from 1-10 while AddedLines can be in the hundreds), which means their contributions are not balanced.
3. Is there a correlation between your complexity metric and task resolution rate? Does the metric actually predict difficulty?
4. The submission does not specify whether containers, prompts, bad patches, reviews, and evaluation scripts will be publicly released. Reproducibility depends critically on these artifacts. Which components do you plan to release?
5. How many instances were rejected during manual validation and for what reasons? What is the inter-annotator agreement if multiple annotators were involved?

---

> ### Author Response · Authors · 2025-11-23
>
> We thank the reviewer for their time and detailed feedback.
>
> ---
>
> ### Analysis of synthetic data used in tasks
>
> >The paper lacks ablations and additional analyses to validate the synthetic data generation pipeline. Since synthetic data is a major component, it may be sensitive to different prompts, introduce leakage, or create unsolvable tasks. Specifically:
> The paper states that code reviews are "informative but do not give away the complete solutions" but provides no rigorous evaluation of review quality, realism, or usefulness.
> No validation is provided for the bad patches, are they realistic failure modes that strong models would produce?
>
> We acknowledge the reviewer’s concerns around analysis of synthetic data. In the updated Section 5.5 we show that including our bad patches helps weed out tests that using only the correct patch could not (as in prior work [1] ). This shows that our bad patches characterise a useful range of edge cases. We have conducted further analysis to categorise bad patches and reviews in Appendix A.
>
>
> | Type | Category            | Count |
> |------|---------------------|-------|
> | A    | Wrong Layer         | 19    |
> | B    | Partial Fix         | 10    |
> | C    | Undisciplined       | 44    |
> | D    | Contract Violation  | 19    |
> | E    | Guard Removal       | 6     |
> | F    | Edge Cases          | 2     |
>
>
>
>
>
>
>
> We see that the mistakes in bad patches are distributed across types such as API contract violations, undisciplined spurious edits or edits made at a too high or too low level of abstraction. These correspond to the sort of errors human developers might make. We find that reviews mostly discuss functional correctness but also some higher level discussions on scope and architecture. We include examples in the Appendix A.
>
> [1] Mündler, N., Mueller, M. N., He, J., & Vechev, M. (2024). SWT-Bench: Testing and Validating Real-World Bug-Fixes with Code Agents [Paper presentation]. The Thirty-eighth Annual Conference on Neural Information Processing Systems. https://openreview.net/forum?id=9Y8zUO11EQ

---

> ### Author Response · Authors · 2025-11-23
>
> ### Lack of evaluation with other models
>
> >The primary evaluation is quite limited. While SWE-Agent and Aider can be considered representative examples of current SWE scaffolding approaches, evaluating only with Gemini 2.5 Flash is insufficient. Comparison with leading open-source models (Qwen3-Coder/GLM-4.x/Kimi-K2) and ideally frontier models (gpt/sonnet) would provide a more comprehensive understanding of how modern LLMs perform on the proposed benchmark.
>
> We have updated the paper with results from DeepSeek-v3.1, GPT-5-mini and Qwen3-32B along with related analysis.
>
> ---
>
> ### Re Appendix
>
> We have added the prompts used for baselines as well as synthetic data generation to Appendix B. Further analysis and experimental details have also been added as detailed in the common response on OpenReview.
>
>
> ---
>
> ### Answers to questions:
>
> > What specific patterns emerge in test generation failures? Are agents failing to understand the bug, unable to construct proper test syntax, or missing edge cases?
>
> We observe many instances where the generated test does not account for edge cases. For example in the test listed in Appendix G, GPT-5 is close but misses the case where the type length is just too long in general. It only tests for excessive nesting. This results in a bad patch succeeding when it should not.
>
> > The formula (ΔFiles + Hunks + AddedLines + RemovedLines)/10 lacks theoretical or empirical justification. Could you explain how this formula was derived? Different components seem to have vastly different scales (e.g., ΔFiles typically ranges from 1-10 while AddedLines can be in the hundreds), which means their contributions are not balanced.
>
> Thank you for raising this point. The formula was mistakenly specified in the paper due to a typo. We have updated the formula to ΔFiles + Hunks + (AddedLines + RemovedLines)/10. Note that only the number of lines modified are divided by 10.
>
>
> > Is there a correlation between your complexity metric and task resolution rate? Does the metric actually predict difficulty?
>
> A very interesting point. To investigate this we plotted the relationship between complexity and bug-fixing success. We have added the results to Section 5.4 and Fig 12 (in Appendix D)  We do indeed see that unresolved patches exhibit higher complexity.
>
> > The submission does not specify whether containers, prompts, bad patches, reviews, and evaluation scripts will be publicly released. Reproducibility depends critically on these artifacts. Which components do you plan to release?
>
> We plan to release all required artifacts for reproduction including prompts, data (bad patches, reviews and other metadata), containers along with the code.
>
> > How many instances were rejected during manual validation and for what reasons? What is the inter-annotator agreement if multiple annotators were involved?
>
> A single validator was used for each python task that was included in the dataset.

---

> > ### Comment · Reviewer_8Gg7 · 2025-11-27
> > **Response**
> >
> > Thank you for your detailed rebuttal and for improving the paper by including additional model evaluations and providing the missing prompts. However, I am keeping my score as is. While the proposal to expand software engineering benchmarks beyond simple bug-fixing is timely and the multi-language support is valuable, the methodology for constructing the benchmark lacks the necessary rigor to serve as a community standard. The reliance on synthetic data for the new task types introduces a dependency on the quality of the generative model that has not been sufficiently validated; the categorization provided in the rebuttal is descriptive rather than evaluative, failing to prove that these artifacts represent realistic engineering challenges. Furthermore, the confirmation that manual validation was performed by a single annotator (specifically for Python, leaving the harder C++/Java subsets ambiguous) raises significant concerns about the dataset's reliability and error rate. Given that benchmarks serve as ground truth for future research, this level of validation is insufficient.

---

### Official Review · Reviewer_SNPc · 2025-11-01

**Soundness:** 2
**Presentation:** 3
**Contribution:** 3
**Rating:** 4
**Confidence:** 4

**Summary:**

OmniCode aims to propose a benchmark for evaluating LLM coding agents by focusing on a range of real-world SE tasks. In particular, the authors consider bug fixing, test generation, code review, and style fixing into a single, manually validated benchmark. However, the benchmark itself suffers from some critical weaknesses: the design for some of the tasks is flawed (discussed below). While OmniCode is a valuable prototype, it currently lacks the rigor to fully assess the nuanced capabilities of coding agents across SE tasks.

**Strengths:**

1. With growing interest and evolving design of coding agents, the setup in OmniCode presents a comprehensive way to evaluate LLM capabilities beyond over-engineered, task-specific solutions.

2. OmniBench allows for vital cross-language analyses, providing sufficient scale and diversity across the four SE tasks.

**Weaknesses:**

I am most not convinced by the design for some of the tasks.
1. Code review: Real-world code reviews often contain discussions centering high-level, systemic reasoning; sometimes performance optimizations; or even code deduplication. By limiting the task to merely "generate instructions" to fix bad code, the benchmark is significantly simplified.

2. Code style: An ideal design should challenge the agent's udnerstanding of idiomatic langauge features and style choices that enhance maintainability and readability. The current setup lacks the depth to measure these aspects, and are now a simple measure of an agent's ability to apply automated linting rules.

3. Test generation is brittle: While the "bad patches" strategy is interesting, its effectiveness relies entirely on the quality, diversity, and plausibility of those incorrect patches. An ideal test should focus on testing boundary conditions or invariants.

**Questions:**

1. Did the authors assess consistency of agent performance across tasks, i.e., does good performance on bug fixing predict success in code review responses?

---

> ### Author Response · Authors · 2025-11-23
>
> We thank the reviewer for their valuable comments.
>
> ---
>
> ### Code Reviews
>
> > Code review: Real-world code reviews often contain discussions centering high-level, systemic reasoning; sometimes performance optimizations; or even code deduplication. By limiting the task to merely "generate instructions" to fix bad code, the benchmark is significantly simplified.
>
> We agree with the reviewer that real world code reviews contain discussion at diverse levels of abstraction. We argue that prompting the LLM to write a code review with the problem statement, correct patch and incorrect patch as context is sufficient to generate a useful distribution of reviews. We have added the review generation prompt used to Appendix B. Performing qualitative analysis of reviews (detail in Appendix A), we find that reviews mostly discuss functional correctness but also some higher level discussions on scope and architecture. Examples are provided in Appendix A.
>
> Note that reviews generated in this way can be used as a starting point for other more specialised tasks (e.g. performance optimisation) that we plan to include in the future. While validation of patches is a difficult problem in these settings, following a similar approach to style review (by using an oracle) is a promising approach.
>
> ---
>
> ### Style Fix task
>
> > Code style: An ideal design should challenge the agent's udnerstanding of idiomatic langauge features and style choices that enhance maintainability and readability. The current setup lacks the depth to measure these aspects, and are now a simple measure of an agent's ability to apply automated linting rules.
>
> We agree with the reviewer that this task should challenge the language models ability to write idiomatic code. To achieve this, we select a subset of rules available in linting tools to prepare our tasks. These correspond to a reasonable set that a developer may care about in day-to-day programming, excluding esoteric rules along with common practices. Example of non-trivial rules that may require non-local changes  -
>
> -  CyclomaticComplexity (Java): Methods or classes that have high code complexity and that should be refactored
> - assignment-from-no-return: Assigning output of functions that do not have a return statement.
> - cppcoreguidelines-pro-type-union-access: Accessing members of unions directly may not be safe.
>
> Details of all rules selected are presented in the Appendix C.2

---

> ### Author Response · Authors · 2025-11-23
>
> ### Test generation
>
> > Code style: An ideal design should challenge the agent's udnerstanding of idiomatic langauge features and style choices that enhance maintainability and readability. The current setup lacks the depth to measure these aspects, and are now a simple measure of an agent's ability to apply automated linting rules.
>
> Prior work in this domain evaluates the test generation abilities of agents by marking a proposed test as correct if it passes on the correct fix. By introducing “bad patches” which need to fail the proposed test, our variant of this task more robustly evaluates tests. These bad patches naturally introduce various edge cases that might arise in real development as shown by the diverse categories of mistakes exhibited (Appendix A).
> For example, consider the difference between the correct patch and the generated bad patch below. The incorrect patch contains the correct guard (returning if either username or password is None) but inserts in in the wrong position.
>
> Correct patch:
>
> ```diff
> diff --git a/django/contrib/auth/backends.py b/django/contrib/auth/backends.py
> --- a/django/contrib/auth/backends.py
> +++ b/django/contrib/auth/backends.py
> @@ -39,6 +39,8 @@ class ModelBackend(BaseBackend):
>      def authenticate(self, request, username=None, password=None, **kwargs):
>          if username is None:
>              username = kwargs.get(UserModel.USERNAME_FIELD)
> +        if username is None or password is None:
> +            return
>          try:
>              user = UserModel._default_manager.get_by_natural_key(username)
>          except UserModel.DoesNotExist:
> ```
>
>
> Bad Patch:
>
> ```
> diff --git a/django/contrib/auth/backends.py b/django/contrib/auth/backends.py
> index ada7461..4087f6e 100644
> --- a/django/contrib/auth/backends.py
> +++ b/django/contrib/auth/backends.py
> @@ -37,6 +37,8 @@ class ModelBackend(BaseBackend):
>      """
>
>      def authenticate(self, request, username=None, password=None, **kwargs):
> +        if username is None or password is None:
> +            return
>          if username is None:
>              username = kwargs.get(UserModel.USERNAME_FIELD)
>          try:
> ```
>
> We specifically discuss the importance of bad patches for the test generation task in Section 5.5. Prior work (SWT-Bench [1]) solely uses the correct patch to evaluate tests. In Section 5.5 we show that including our bad patches helps weed out tests that the correct patch could not (e.g. g., Qwen C++ would be 22.7% instead of 4.55%, Qwen Java would be 7.79% instead of 1.3%, DeepSeek C++ would be 43.8% instead of 25%, and DeepSeek Java would be 28.4% instead of 11.9%) This shows that our bad patches characterise a useful range of edge cases.
>
> [1] Mündler, N., Mueller, M. N., He, J., & Vechev, M. (2024). SWT-Bench: Testing and Validating Real-World Bug-Fixes with Code Agents [Paper presentation]. The Thirty-eighth Annual Conference on Neural Information Processing Systems. https://openreview.net/forum?id=9Y8zUO11EQ

---

> > ### Author Response · Authors · 2025-11-23
> >
> > > Did the authors assess consistency of agent performance across tasks, i.e., does good performance on bug fixing predict success in code review responses?
> >
> > Thanks for this interesting question. We are better able to evaluate consistency of agents / models across tasks due to our extended evaluation in the updated paper. We find that when using SWE-Agent, performance of different models on bug-fixing is strongly correlated to review-response (pearson coeff = 0.921) and weakly correlated to test generation (pearson coeff = 0.764). We find the correlation does not hold for style-review however (perason coeff = 0.512), where Gemini-2.5-Flash performs as good as or better than DeepSeek v3.1 on Style-Fix despite DeepSeek consistently outperforming Gemini on Bug-Fix. We find these observations to be generally true for Aider too, albeit slightly weaker. Section 5.1 has been updated with this observation along with details in Appendix E.

---

### Official Review · Reviewer_Bh5q · 2025-11-01

**Soundness:** 2
**Presentation:** 3
**Contribution:** 2
**Rating:** 2
**Confidence:** 5

**Summary:**

The authors propose OmniCode a code agents benchmark combining instances from SWE-Bench and Multi-SWE-Bench with recently mined new instances. The data sets are enhanced by using LLMs to create new task types from existing instances. More precisely, the authors add three tasks to the standard "issue resolving" task: 1) Test generation, 2) responding to code review, and 3) code style application. For 2) and 3) an LLM is used to create bad patches which are related to the ground truth patch but do not solve the task at hand. Any test generated for the test generation task has to fail for the bad patches and pass for the ground truth patch. Bad patches are also used to create "code reviews" where an LLM is tasked to generate a review of a bad patch with knowledge about the ground truth. The task is then to "respond" to the review to fix it and arrive at the ground truth solution. In the experimental section, the authors compare both the Aider and SWE-agent scaffold with Gemini 2.5 Flash and demonstrate varying performance over the different tasks.

**Strengths:**

* OmniCode combines multiple important tasks over multiple languages. In particular the latter is important but oftentimes overlooked. I'd love to see the authors to expand the supported languages (e.g., with JavaScript and TypeScript instances from other benchmarks that offer verified splits).
* The manuscript is well written and easy to understand. Visualizations clearly convey the key results and experimental findings, making it straightforward to follow the authors' analysis and conclusions.

**Weaknesses:**

* Bad patch generation: A bad patch is defined as one that doesn't pass the golden tests. If the golden tests are too narrow or too permissive bad patches may reflect these shortcomings and the code review task would be affected by this as well.
* Quality and solvability. LLMs are used for patch and review generation. Especially since the latter builds on top of the first LLM results, the risk for decreased quality and potential impacts on solvability multiplies (LLMs on LLMs). In general it is of limited usefulness for the community to develop benchmarks based on LLM-generated inputs. If we want the agents to support humans, inputs should come from humans. Despite being trained on human preference data, typical LLMs will not write the same patches or review messages as an SDE or VibeCoder. On a larger scale this may eventually harm the field as we measure performance of coding agents on inputs that are not from the same distribution in which we'd like to use them.

**Questions:**

* You are already combining instances from SWE-Bench and Multi-SWE-Bench, is there a reason why you don't add instances from another verified multi-language data set like SWE-PolyBench[1]?
* Did you verify that the test cases for a given could lead to false positives (or even false negatives)?

1. Rashid, M. S., Bock, C., Zhuang, Y., Buchholz, A., Esler, T., Valentin, S., ... & Callot, L. (2025). SWE-PolyBench: A multi-language benchmark for repository level evaluation of coding agents. arXiv preprint arXiv:2504.08703.

---

> ### Author Response · Authors · 2025-11-23
>
> We thank the reviewer for their valuable feedback.
>
> ---
>
> ### Bad patch generation
>
> >Bad patch generation: A bad patch is defined as one that doesn't pass the golden tests. If the golden tests are too narrow or too permissive bad patches may reflect these shortcomings and the code review task would be affected by this as well.
>
> We rely on the assumption that current tests in the repository correctly specify some required behaviour. This is independent of the coverage / breadth / specificity of the tests. As long as a patch fails one of these tests (as in the case of bad patches), we conclude that it violates this specification in some way and hence is an incorrect fix.
>
> ---
>
>
> ### Use of synthetic data
>
> While we understand the reviewers concerns, we believe that synthetic data can play an important role in benchmark creation if used carefully. Bad patches and reviews are difficult to find in a clean form in real world data and creating them manually is prohibitively expensive. Using LLMs is a scalable and inexpensive way to do this.
>
> In the updated Appendix A, we show that our synthetic data falls broadly into the categories of mistakes and reviews that humans would exhibit. We provide examples to illustrate this.
>
> In Section 5.5, we further see that using the generated bad patches in the test generation task helps filter out tests that would be accepted if only using the gold patch. This shows that the generated bad patches characterise various edge cases.
>
> We believe this is a useful contribution to the field but it does warrant future work studying the impact of evaluating and training agents on such synthetic data.
>
> ---
>
> ### Answers to questions:
>
> > You are already combining instances from SWE-Bench and Multi-SWE-Bench, is there a reason why you don't add instances from another verified multi-language data set like SWE-PolyBench[1]?
>
> 1. Thank you for sharing SWE-PolyBench, we have built OmniCode with Multi-SWE-Bench as it was available when we started building the benchmark. Given the general nature of OmniCode, it is very much feasible to onboard more tasks from SWE-PolyBench and we hope to take on such expansion efforts in the future.
>
>
> > Did you verify that the test cases for a given could lead to false positives (or even false negatives)?
>
> 2. The tasks for Java and C++ are sourced from Multi-SWE-Bench which employs a manual verification procedure to ensure that tests and task descriptions are aligned. We perform similar validation for the Python tasks.

---

> > ### Comment · Reviewer_Bh5q · 2025-11-23
> > **Thank you**
> >
> > Thank you for providing your point of view on synthetically generated data for benchmarking. While I don't have problems with using such data for training purposes, I vehemently oppose the introduction of LLM-generated benchmarks as they do not represent the target distribution and therefore do not measure what they claim. Instead, you're measuring the performance of systems if used by LLMs. As long as our systems are used by humans, only human-generated data can be reliably serve as proxy for future performance. I will keep my score as is.

---

### Official Review · Reviewer_S6pD · 2025-11-01

**Soundness:** 2
**Presentation:** 1
**Contribution:** 2
**Rating:** 2
**Confidence:** 5

**Summary:**

The paper introduces OmniCode, a new benchmark designed to evaluate LLM-powered software development agents beyond the narrow scope of existing benchmarks like HumanEval and SWE-Bench. The authors argue that real-world software engineering involves a more diverse set of tasks. OmniCode addresses this gap by providing 1,794 tasks across three programming languages (Python, Java, C++) and four key task categories: bug fixing, test generation, responding to code reviews, and fixing style violations.

However, the paper's writing looks incomplete. The appendix section still has placeholder text instead of actual content. I also didn't see any specific details in the supplementary materials.

**Strengths:**

- The paper addresses a clear and widely recognized gap in the field. Current benchmarks focus heavily on bug fixing or single-function generation. OmniCode provides a much-needed holistic benchmark that covers a wider, more realistic spectrum of the software development lifecycle, including testing, code review, and style adherence

- The paper provides a strong set of baseline experiments. The results are insightful, such as the clear identification of Test Generation as a major weakness for current agents and the nuanced finding that code reviews help on complex Java/C++ tasks but may hurt performance on simpler Python tasks.

-

**Weaknesses:**

-  The paper's writing looks incomplete. The appendix section still has placeholder text instead of actual content. I also didn't see any specific details in the supplementary materials.

- The robustness of the "Test Generation" and "Code Review" tasks hinges on the quality of the synthetic "bad patches" and "review reports." The paper details how these are generated (e.g., using weaker agents or LLM-based perturbation), but a more in-depth qualitative analysis of their diversity and realism would strengthen the paper. For instance, how do we know the "bad patches" cover a truly diverse set of realistic human errors?

- The evaluation for the "Code Style" task focuses on quantifying the reduction of linter-reported issues using a specific score. However, it is not explicitly stated whether the project's functional test suite is run after the style fix. A good style fix should not introduce functional regressions, and this would be a valuable check to include for a more robust evaluation.

**Questions:**

Regarding the "Code Style" task: Did the evaluation process involve running the functional test suite after an agent applied a style fix? It seems critical to verify that the agent did not introduce functional regressions while refactoring the code to resolve style violations.

---

> ### Author Response · Authors · 2025-11-23
>
> We thank the reviewer for their time and valuable comments.
>
> ---
>
> ### Re: Appendix
>
> We have updated Appendix B with prompts used for baselines as well as synthetic data generation.
>
> ---
>
> ### Analysis of synthetic data
>
> >The evaluation for the "Code Style" task focuses on quantifying the reduction of linter-reported issues using a specific score. However, it is not explicitly stated whether the project's functional test suite is run after the style fix. A good style fix should not introduce functional regressions, and this would be a valuable check to include for a more robust evaluation.
>
>
> We acknowledge the reviewer’s concerns around analysis of synthetic data. We have conducted further analysis to categorise bad patches with details in Appendix A. We see that the types of mistakes in bad patches are distributed across types such as edits at the wrong abstraction layer, API contract violations and undisciplined spurious edits. These correspond to the sort of errors human developers might make. We also include examples in the Appendix A.
>
> ---
>
> ### Functionality testing with style-fixing
>
>
> >The robustness of the "Test Generation" and "Code Review" tasks hinges on the quality of the synthetic "bad patches" and "review reports." The paper details how these are generated (e.g., using weaker agents or LLM-based perturbation), but a more in-depth qualitative analysis of their diversity and realism would strengthen the paper. For instance, how do we know the "bad patches" cover a truly diverse set of realistic human errors?
>
> We agree with the reviewer that it is important to ensure that the agent's attempts at style-fixing do not alter the functionality of the code. To do this we have run the test suites on the results of style-review evaluations for Python. Assigning a score of 0 to instances that fail results in the following -
>
> | Model             | Before | After |
> |-------------------|--------|--------|
> | Gemini-2.5-Flash  | 72.2%  | 57.0% |
> | DeepSeek-V3.1     | 73.4%  | 54.0% |
> | GPT-5-mini        | 56.3%  | 45.9% |
> | Qwen3-32B         | 22.7%  | 19.5% |
>
> We find that there is an average drop of ~10% for the scores. We are in the process of running tests suites for Java and C++ tasks. These will be updated in the main text.

---

### Author Response · Authors · 2025-11-23
**Response to common concerns**

We thank the reviewers for their comments and suggestions. They have recognised that OmniCode fills an important gap in the current SWE-Agent benchmarking landscape along with the strong set of evaluations.  We have updated the manuscript with the following major changes taking into account their feedback -

1. Evaluation of three additional SOTA models: DeepSeek-v3.1, GPT-5-mini and Qwen3-32B

| Language | Model            | Bug-Fixing | Test-Generation | Review-Response | Style-Fixing |
|----------|------------------|------------|----------------|-----------------|-------------|
| **Python** | Gemini-2.5-Flash | 38.1 %     | 14.0 %         | 29.9 %          | 72.2 %      |
|          | DeepSeek-V3.1     | 56.4 %     | 18.7 %         | 52.2 %          | 73.4 %      |
|          | GPT-5-mini        | 47.3 %     | 6.2 %          | 30.5 %          | 56.3 %      |
|          | Qwen3-32B         | 24.5 %     | 4.0 %          | 17.7 %          | 22.7 %      |
| **C++**   | Gemini-2.5-Flash | 8.0 %      | 12.2 %         | 13.6 %          | 36.3 %      |
|          | DeepSeek-V3.1     | 19.6 %     | 25.0 %         | 22.7 %          | 30.2 %      |
|          | GPT-5-mini        | 15.2 %     | 6.8 %          | 20.5 %          | 21.8 %      |
|          | Qwen3-32B         | 3.8 %      | 4.5 %          | 4.5 %           | 8.6 %       |
| **Java**  | Gemini-2.5-Flash | 14.7 %     | 4.9 %          | 31.6 %          | 60.4 %      |
|          | DeepSeek-V3.1     | 31.2 %     | 20.9 %         | 44.3 %          | 50.2 %      |
|          | GPT-5-mini        | 22.0 %     | 2.7 %          | 26.6 %          | 25.0 %      |
|          | Qwen3-32B         | 10.1 %     | 1.3 %          | 15.2 %          | 23.3 %      |


2. Analysis of synthetic bad patches and reviews (details in Appendix A). We characterise bad patches and reviews into various categories demonstrating their diversity and providing examples.


3. Additional analysis -

    - Section 5.5: Importance of bad patches. We show that including bad patches when evaluating model generates tests reduces false positives (tests incorrectly evaluated as correct).

    - Section 5.4: We observe that unresolved instances tend of have higher gold patch complexity. Complexity analysis of generated patches shows that unresolved patches usually exhibit "explosive" complexity as agents attempt sprawling refactors when unable to find a precise fix.

    - Section 5.3: Further analysis of Review-Response results showing that including reviews consistently improves agent’s ability to resolve issues.

    - Section 5.2: Cross-Agent Analysis : Expanded on the difference in results between aider and sergeant across tasks.


4. Additions to appendix -
    - Appendix A: Analysis of synthetic bad patches and reviews.
    - Appendix B: Prompts used for synthetic data creation and baselines in the appendix
    - Appendix C: Style Review
        - Style review results for Python after running test suite to ensure patches maintain functionality
        - Analysis showing alternate ways of quantifying Style Review behaviour with Fix Rate (% of original errors resolved) and Error Ratio  (errors after / errors before)
        - Linting rulesets selected for Style Review Task.
        - Rule-wise analysis of results
        - Rate of successfully generating correct patches .
    - Appendix D: Analysis of correlation between patch complexity and instance resolution
    - Appendix E: Analysis of correlation between performance across tasks
    - Appendix F: Patch generation rate across tasks.

5. Complete code and data will be released.

The additions to the manuscript have been marked in blue.

---

### Note · Authors · 2026-01-05

I have read and agree with the venue's withdrawal policy on behalf of myself and my co-authors.